help-seeking; mental health; help-seeking behaviors; help-seeking attitudes; help-seeking intentions

**Corresponding author:**
Abhijit Nadkarni;
Email: abhijit.nadkarni@lshtm.ac.uk

# Interventions to improve mental health help-seeking attitudes, intentions and behaviors: A systematic review of recent advances

Shruti Bora[1] , Bijayalaxmi Biswal[1], Yashi Gandhi[1,2], Sloka Iyengar[1],
Richard Velleman[1,3], Daisy Radha Singla[4] , Vikram Patel[5] ,
Marimilha Grace Pacheco[1], Nikita Nalawade[6] and Abhijit Nadkarni[1,2]

[1]Addictions and Related Research Group, Sangath, India; [2]Epidemiology and Population Health, London School of Hygiene & Tropical Medicine, UK; [3]Department of Psychology, University of Bath, UK; [4]Campbell Family Mental Health Research Institute, Centre for Addiction and Mental Health, Canada; [5]Department of Global Health and Social Medicine, Harvard Medical School, USA and [6]Queen Mary University of London, UK

## Abstract

The gap between mental health conditions and care uptake remains a global challenge, despite the availability of effective and affordable treatments. This gap is driven by demand-side barriers, such as lack of mental health literacy (MHL), stigma, etc., that hinder help-seeking. In this systematic review, we critically appraise interventions aimed at promoting help-seeking for mental health conditions. The review protocol was prospectively registered with PROSPERO (registration number CRD42021273843). A systematic search was conducted across MEDLINE, PsycINFO, Embase, Global Health, Cumulative Index to Nursing and Allied Health Literature (CINAHL) and Cochrane Central Register of Controlled Trials (CENTRAL). Only RCTs published after 2016, testing interventions with the aim of improving help-seeking behaviors, intentions and attitudes for any mental health conditions, were included. Due to the heterogeneity of outcomes and measures used in the studies, a narrative synthesis was conducted to examine the evidence. Fifty-four studies met the inclusion criteria. Our review confirms that MHL or psychoeducation, motivational interviewing (MI) and social contact interventions effectively improve help-seeking attitudes ($n = 10$), intentions ($n = 17$) and behaviors ($n = 16$). Multi-component MHL and MI-based strategies enhance help-seeking behaviors, while social contact online interventions enhance intentions. MHL/psychoeducation was effective across all outcomes, particularly when combined with other strategies. Despite a rise in help-seeking research, many studies lacked standardized frameworks, making cross-intervention comparisons difficult. Future work should align with theoretical models of help-seeking and explore mechanisms of change to better understand the link between intentions, attitudes and behaviors.

## Impact statement

Although effective and affordable treatments exist, many people continue to face barriers that prevent them from seeking or receiving help. These barriers include not only the lack of available services and high treatment costs but also fear of stigma, low awareness about mental health and doubts about the usefulness of treatment. Addressing these "demand-side" factors is therefore as essential as expanding services. Our review focuses on this critical challenge. How to encourage people to recognize their need for support and take active steps toward seeking help? Help-seeking is not a single event but a process that involves understanding one's symptoms, deciding to seek support and following through with that decision. We examined recent evidence on interventions that aim to influence three key components: attitudes, intentions and behaviors. We found a growing body of work in this area, with over 50 new studies published in the past few years. Interventions that combined multiple strategies, such as psychoeducation, stigma reduction, motivational approaches and promotion of available resources, were the most effective. Social contact-based approaches, particularly those delivered through short videos, showed consistent promise in improving help-seeking attitudes and intentions, especially among the general population. However, translating improved attitudes and intentions into actual behavior change remains a challenge. Our findings highlight an important shift in mental health research from focusing primarily on service delivery to understanding how people decide to seek help. By synthesizing recent evidence, this review contributes to a clearer understanding of what works to encourage help-seeking and where the gaps remain. It underscores the need for future research to use theory-driven frameworks, examine mechanisms of change and adopt standardized measures. Focusing on how interventions can effectively move people from awareness to action is essential for reducing the global mental health treatment gap and ensuring that more individuals receive appropriate care.

## Introduction

The disparity between the prevalence of mental health conditions and the uptake of mental health care remains high globally, despite the availability of effective and affordable treatments (Kohn et al., 2004; Codony et al., 2009; White et al., 2018). Barriers perpetuating this treatment gap include both demand-side factors (e.g., perceived lack of confidentiality, trust issues with providers, stigma, low mental health literacy, etc.) and supply-side factors (e.g., lack of mental health care services, insufficient public funding, high out-of-pocket costs, etc.) (Gulliver et al., 2010; Luitel et al., 2017; Wainberg et al., 2017; Qin and Hsieh, 2020; Babatunde et al., 2021; Sarikhani et al., 2021). While much research has focused on reducing the treatment gap by addressing supply-side barriers (e.g., using approaches such as task sharing), increasing the availability of mental health services alone does not guarantee greater service uptake (Roberts et al., 2022). There is now a growing interest in addressing demand-side barriers by developing interventions to promote help-seeking for mental health conditions. However, encouraging help-seeking remains a significant challenge for public health (Schnyder et al., 2017; Organization, World Health, 2004a, 2004b). Several factors deter individuals from help-seeking for mental health conditions, including low perceived need for treatment or support, low MHL, doubts about the perceived benefits of help seeking (ten Have et al., 2010; Rughani et al., 2011), financial constraints and stigma (Sanders Thompson et al., 2004; Andrade et al., 2014; Topkaya, 2015; Umubyeyi et al., 2016; Schnyder et al., 2017; McLaren et al., 2023).

Help-seeking is a dynamic process that begins with recognizing mental health symptoms, followed by expressing the need for support, identifying available help, and ultimately seeking help (Rickwood et al., 2005; Xu et al., 2018) but lacks consensus on definition or measurement, often varying by process stage or help source (professional versus non-professional) (Rickwood and Thomas, 2012). The Theory of Planned Behavior (TPB) has gained substantial attention over the years in understanding help-seeking, focusing on three processes (Ajzen, 1991; Adams et al., 2022; Naumova, 2022): help-seeking attitudes (overall perception toward seeking help), intentions (plans/decisions to seek help), and behaviors (actually seeking help/support) (Rickwood and Thomas, 2012). TPB suggests that attitudes influence intentions, which predict behaviors (Ajzen, 1991). Research confirms that TPB effectively predicts intentions and behaviors (McEachan2011; Ajzen, 1991; Armitage and Conner, 2001) and that intentions mediate the relationship between attitudes and behaviors (Kim and Hunter, 1993; Glasman and Albarracín, 2006). Rickwood and Thomas (2012) further advocated for a TPB-based framework to conceptualize and measure help-seeking as a complex process and all three components are crucial for influencing the help-seeking process comprehensively. Following the TPB model, our review focuses on attitudes, intentions and behaviors as primary outcomes because each serves a distinct public health function. Improving attitudes and intentions among the general population is critical for long-term, population-level stigma reduction. Conversely, targeting intentions and actual behaviors is essential for individuals currently experiencing mental health conditions to ensure they transition from recognizing a need to seeking help. By including all populations and contexts, this review comprehensively captures the full spectrum of demand-side strategies necessary to address the treatment gap globally (Rickwood and Thomas, 2012; Adams et al., 2022).

The effectiveness of interventions aimed at improving help-seeking for mental health conditions remains unclear, as evidenced by mixed results in previous literature (Gulliver et al., 2012; Adams et al., 2022). Considering the changing landscape of mental health care and the gaps in current research, our review synthesizes the recent evidence on effective interventions for promoting help-seeking processes (attitudes, intentions and behaviors) for mental health conditions. We synthesize post-2016 RCTs on these outcomes, focusing on recent studies not covered by Xu et al. (2018), who synthesized RCTs up to 2016 using meta-analysis. Our narrative synthesis approach differs substantially due to greater heterogeneity in post-2016 studies, precluding a meta-analysis. As noted by Rickwood and Thomas (2012), studies often fail to specify which part of the help-seeking process (e.g., recognition versus action) or which source of help (e.g., formal versus informal) is being targeted. This lack of standardization justifies our broad inclusion criteria, as it allows for a comprehensive mapping of diverse interventions ranging from MHL to social contact, addressing different stages of the process.

Our review aimed to identify and evaluate strategies that effectively improve help-seeking behaviors, attitudes and intentions. We also synthesized MHL, stigma and related mediators as secondary outcomes due to their complex relationship with help-seeking (Jorm et al., 1997; Smith et al., 2008; Schnyder et al., 2017; Cheng et al., 2018; Michel et al., 2018; Mackenzie et al., 2019; McLaren et al., 2023).

## Methods

The protocol for the review was registered on PROSPERO (CRD42021273843; https://www.crd.york.ac.uk/PROSPERO/view/CRD42021273843) (Nadkarni and Gandhi, 2021). This review is developed and reported according to the Preferred Reporting Items for Systematic Reviews and Meta-Analyses (PRISMA) checklist (Page et al., 2021).

### Search strategy

Our search was conducted on six databases: Ovid Medline, EMBASE, PsycINFO, Cochrane Central Register of Controlled Trials (CENTRAL), Global Health and Cumulative Index to Nursing and Allied Health Literature (CINAHL). The initial search was in January 2022, updated in May 2023 and focused on "mental health," "help-seeking" and "randomized controlled trials." The detailed search strategy is in Supplementary Appendix A.

### Eligibility criteria

Our review included randomized controlled trials (RCTs) from any country or setting, with populations either non-clinical (general population) or clinical (individuals with mental health conditions, as defined by ICD, DSM, clinical diagnosis, standardized questionnaires or self-defined psychological issues). There were no age restrictions, and only English publications after December 2016 were considered to capture recent evidence.

Interventions were eligible if they aimed to promote help-seeking for mental health care, with at least one help-seeking outcome (attitudes, intentions or behaviors) as a primary outcome. Controls could be active or inactive. Reviews, meta-analyses, commentaries, opinion pieces, pilot studies, observational studies and non-randomized trials were excluded.

## Study selection and data collection

After database search completion, the data were imported to Endnote (The EndNote Team, 2013), with duplicates removed automatically and manually. Records were screened in Covidence (Anon, 2023) by two pairs of reviewers (SI and MGP, MGP and SB) for titles and abstracts, and full texts were independently reviewed by two pairs (NN and MGP, MGP and SB), with reasons for exclusion recorded. Inter-rater reliability showed a Kappa of 0.50 (NN and MGP) and 0.89 (SB and MGP). Disagreements were resolved by a third reviewer (AG or YG). Data extraction was done independently by reviewers (NN, MGP, NH and SB) and reviewed by SB.

## Data

We extracted data on help-seeking outcomes as behaviors (self-reported/observable service use, e.g., appointments/helplines), intentions (plans/decisions to seek help) and attitudes (beliefs/willingness) (Rickwood and Thomas, 2012). We also extracted information regarding the type of help accessed (e.g., formal professional services, informal peer support or self-help), categorized the outcome measures by their source (e.g., administrative records, clinical registries or self-report scales) and recorded the specific timeframes for each outcome, which ranged from immediate post-intervention effects to 12-month or 24-month longitudinal follow-ups and lifetime help-seeking history.

While the Attitudes Toward Seeking Professional Psychological Help Scale (ATSPPH) is primarily a measure of attitudes, several included studies (e.g., Amsalem et al., 2022; Amsalem, Wall et al., 2023) utilized specific items from the scale, such as "openness to treatment seeking" to operationalize help-seeking intentions. We have categorized these results under "intentions" to remain consistent with the authors' primary research objectives.

Additional data extracted included information about:

a) MHL (knowledge about mental health problems) (Jorm et al., 1997; Xu et al., 2018);
b) mental illness stigma, including perceived and personal stigma (Xu et al., 2018);
c) factors mediating the effect of the help-seeking intervention (Xu et al., 2018).

We initially planned to examine implementation outcomes like feasibility, acceptability and appropriateness, but decided against it due to data heterogeneity and the limited number of studies reporting these outcomes.

## Analysis and data synthesis

While Xu et al. (2018) utilized meta-analysis, the post-2016 literature reflects a significant diversification in modalities (e.g., social media vs. standalone apps) and a shift toward multi-component interventions. We determined that the clinical and methodological heterogeneity, specifically the lack of standardized follow-up intervals and the variety of help-seeking sources (formal vs. informal), would render a pooled effect size misleading. Given the heterogeneity in the interventions and outcome measurement, a descriptive analysis of the included studies is presented in a narrative format (Popay et al., 2006). An intervention was considered effective if it showed significant differences in help-seeking outcomes ($p \leq 0.05$ or $p \leq 0.01$) compared to the control group; otherwise, it was deemed ineffective. Where reported by primary studies, standardized effect size estimates (e.g., odds ratios, Cohen's $d$, partial $\eta^2$) were extracted and presented

alongside statistical significance to aid interpretation of intervention magnitude (see Table 3). These values are presented descriptively to provide context for the reported statistical significance; they are not intended for direct cross-study comparison.

Based on the primary outcomes, we categorized studies into three groups: (a) behaviors, (b) intentions and (c) attitudes. We further categorized interventions based on their delivery settings within the larger groups: school/university-based, facility-based and community-based.

To synthesize the diverse intervention strategies identified, we employed an inductive, iterative approach to categorize intervention components based on their frequency and functional similarity. Given that a majority of the included studies utilized multi-component interventions such as combining psychoeducation with social contact or MI, arranging the results by individual study would have obscured broader research trends. Therefore, we grouped strategies into thematic categories (e.g., "Media-based," "Social Contact" or "MI Strategies") focused on identifying functional clusters of strategies to provide a clearer and usable framework for implementation. This mapping process (Table 4), allows for a more practical interpretation of which "active ingredients" are most frequently combined to drive changes in help-seeking attitudes, intentions and behaviors. By focusing on these clusters of strategies, the review highlights intervention formats that are most likely to be feasible in real-world settings.

## Risk of bias assessment

The risk of bias was independently assessed by two pairs of three researchers (NN & MGP and MGP & SB). We used the Cochrane Collaboration "Risk of Bias 2" tool (ROB2), which consists of seven domains (sequence generation, allocation concealment, blinding of participants, personnel and outcome assessors, incomplete outcome data, selective outcome reporting and other sources of bias) (Sterne et al., 2019). In case of a difference in assigned scores, the pairs met to review and arrived at a consensus.

## Results

Figure 1 outlines the number of studies included and excluded at each stage. A total of 24,619 records were identified of which 10,854 records were duplicates. Of the remaining 13,765 records, 209 were screened for full-text eligibility, and 54 studies were included as they met the inclusion criteria. Out of 209 full-text articles assessed, the majority (154; 73.7%) were excluded due to an ineligible study design (non-RCTs).

## Study characteristics

The included studies ($n = 54$) comprised a total sample size of 679,652 participants (range: 24 to 333,596). Most of the studies were individual RCTs ($n = 44$), and the rest were cluster RCTs ($n = 10$). Apart from three studies, which were stepped-wedge RCTs (Parikh et al., 2021; Ries et al., 2022; Lee et al., 2023), all were parallel RCTs. Of the 54 included studies, the vast majority (94.4%; $n = 51$) were conducted in High-Income Countries (HICs), with the USA (51.8%; $n = 28$) and Australia (18.5%; $n = 10$) being the most frequent. Only 5.5% ($n = 3$) were conducted in lower-middle income countries (India and Nepal) or upper-middle income countries (China) (see Table 1).

The 54 included studies involved a total of 679,652 participants. The demographic breakdown (see Table 1) is summarized below.

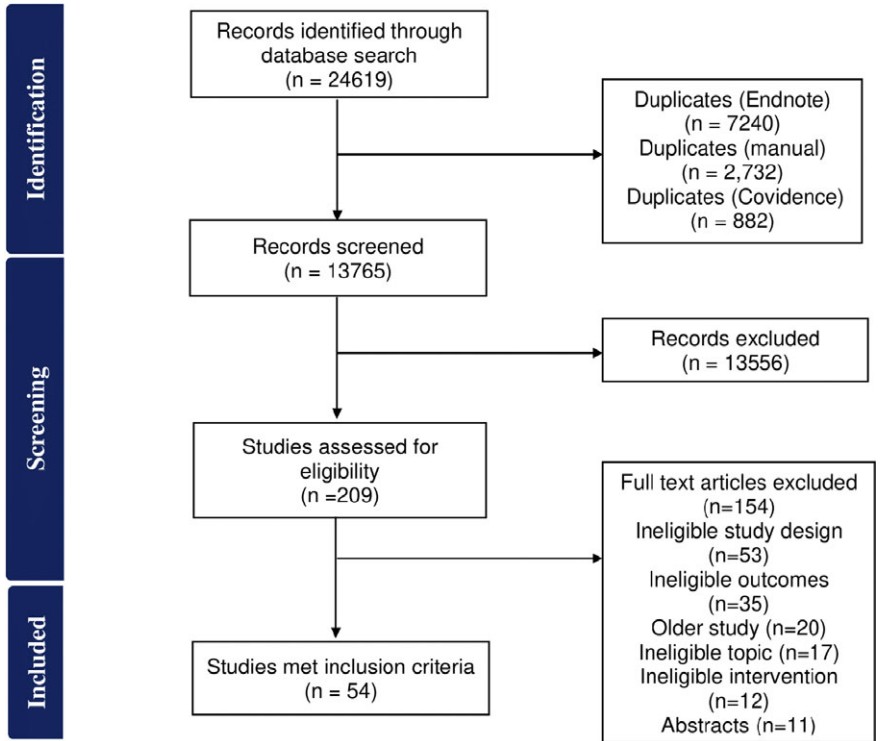

**Figure 1.** Flowchart for study identification, screening, and inclusion.

The majority of studies focused on adults (74.1%; *n* = 40), with a mean age range of 18–73 years. Children and adolescents were the focus of 18.5% (*n* = 10) of studies (mean age 11.5–16.8 years), while 7.4% (*n* = 4) targeted both adults and children/ adolescents (mean age = 12–30.6 years) populations.

Approximately 63.0% (*n* = 34) of the studies were conducted with the general population, while the remaining 37.0% (*n* = 20) targeted clinical or at-risk groups, such as those with existing mental health conditions.

Most studies included both genders. However, four used single-gender samples- one all-female study (Holt et al., 2017) and three all-male studies (King et al., 2018; Nickerson et al., 2020; Achter-bergh et al., 2021).

A portion of the research (*n* = 10) focused on specialized populations, including at-risk veterans (*n* = 3) (Ilgen et al., 2022; Possemato et al., 2023; Stecker et al., 2023); healthcare workers (*n* = 2) (Amsalem, Lazarov, et al., 2022; Amsalem, Wall, et al.,

2023), sexual and gender Minorities (SGM) (*n* = 2) (Achterbergh et al., 2021; Han et al., 2023) and one study each in refugees, people experiencing homelessness and women experiencing postpartum stress (Holt et al., 2017; Nickerson et al., 2020; Kerman et al., 2023).

### Interventions

The 54 included studies targeted various combinations of help-seeking outcomes (see Table 2). Nearly half of the studies targeted help-seeking behaviors exclusively (*n* = 24). Intentions were the sole focus of 15 studies, while only four focused on attitudes.

Eleven studies used an integrated approach targeting multiple outcomes. Six targeted attitudes and intentions (Hussain and Alha-bash, 2020, 2022; Seidman et al., 2022; Tay, 2022; Han et al., 2023; Hollar and Siegel, 2023). Three targeted intentions and behaviors (Coleman et al., 2019; Nickerson et al., 2020; Calear et al., 2022), and two targeted all three outcomes: 3.7% (*n* = 2) (Martin et al., 2020; Tobias et al., 2022).

Across the entire review, 66.7% (*n* = 36) of the tested interventions were effective in significantly improving at least one primary help-seeking outcome. Detailed characteristics for each individual study, including the specific country and the type of help sought, are provided in Table 3.

### Mapping of intervention components

To identify the "active ingredients" of these interventions, we grouped components based on their functional similarity and frequency. This mapping, which cross-references strategies with outcomes and the Risk of Bias (RoB 2) assessment, is presented in Table 4.

Following our methodology, interventions were categorized by primary outcomes, delivery settings and where possible, component approaches. This synthesis highlights "promising"

**Table 1.** Demographic and geographic characteristics of included studies

| Characteristic | Category | *n* (%) |
|---|---|---|
| Age group | Only adults | 40 (74.1%) |
| | Only children/ adolescents | 10 (18.5%) |
| | Mixed (adults + children/ adolescents) | 4 (7.4%) |
| Target population | General population | 34 (63.0%) |
| | Clinical or at-risk population | 20 (37.0%) |
| Geography (setting) | High-income countries (HICs) | 51 (94.4%) |
| | Low- and middle-income countries (LMICs) | 3 (5.6%) |

**Table 2.** Summary of help-seeking outcome targets

| Primary outcome(s) | Total studies (n) | Percentage (%) | Effective studies (n) | Proportion of effective studies (%) |
|---|---|---|---|---|
| Single outcome | | | | |
| Help-seeking behaviors | 24 | 44.40 | 12 | 50.00 |
| Help-seeking intentions | 15 | 27.80 | 12 | 80.00 |
| Help-seeking attitudes | 4 | 7.40 | 3 | 75.00 |
| Combined outcomes | | | | |
| Attitudes + Intentions | 6 | 11.10 | 5 | 83.30 |
| Intentions + Behaviors | 3 | 5.60 | 2 | 66.70 |
| Attitudes + Intentions + Behaviors | 2 | 3.70 | 2 | 100.00% |
| Total | 54 | 100 | 36 | 66.70% |

interventions defined as those achieving statistical significance ($p \leq 0.05$) for effectiveness and demonstrating a notable magnitude of effect (within their specific contexts), such as large odds ratios or effect sizes.

### Interventions targeting help-seeking behaviors

Of the 29 studies targeting behavior, interventions in 16 (55.2%) were effective.

### University/school settings (n = 6/12 effective studies)

Identification-based pathways: Proactive identification strategies proved efficient at converting problem recognition into service initiation. Automated universal screening (Student Assistance Program, SAP) among high school students ($n = 12,909$) increased service initiation fourfold (95% CI 2.0–7.9) by directly linking positive screens to SAP-recommended school or community-based services (Sekhar et al., 2022). Similarly, interactive avatar-based gatekeeper training for undergraduates ($n = 117$) that trained peers to recognize struggling students and direct them toward campus resources significantly increased help-seeking at campus counseling centers (OR = 2.3, $p < .001$) (Coleman et al., 2019). In India, a single-session, lay counselor-led classroom sensitization program for adolescents ($n = 3,587$) achieved an increase in school counseling referrals (OR = 111.36, 95% CI 35.56–348.77, $p < .001$) (Parikh et al., 2021), eliminating access barriers, especially in a resource-constrained setting.

Knowledge with stigma reduction focus: College athletes receiving multi-component customized mental health literacy (athlete-focused) combined with social-contact-based stigma reduction videos and resource promotion ($n = 107$) showed help-seeking increases (OR = 5.06, $p = 0.01$) (Martin et al., 2020) with the intervention explicitly including role-play exercises where athletes developed personalized help-seeking plans, converting knowledge into concrete action steps. School students receiving a 3-module,

3-h anti-stigma curriculum (didactic teaching, group discussion, homework on stigma/mental disorders) increased formal treatment-seeking when baseline symptom levels were high (OR = 3.90, 95% CI 1.09–13.87, $p < .05$) (Link et al., 2020).

Peer-delivered motivation and support: Undergraduate peer counselors trained in motivational interviewing (MI) who delivered a single psychoeducation + MI session to fellow students with needle anxiety ($n = 61$) significantly increased treatment-seeking behaviors (IR = 2.41, 95% CI 1.29–4.50, $p = 0.006$) (Finitsis et al., 2022).

### Healthcare and facility-based settings (n = 4/10 effective studies)

In healthcare settings, high-intensity, clinician-delivered strategies outperformed passive referrals. Linkage models, where dedicated staff use MI strategies to navigate patients from identification to specialized care, demonstrated strong effects among high-risk clinical populations. Primary care linkage workers delivered SBIRT + MI-based case management (Recovery Management Checkups) for SUD patients ($n = 266$, mean age 48.3 years), resulting in 4.59-fold higher treatment initiation overall ($p = 0.001$); and more specifically for residential care (AOR = 3.31, 95% CI 1.28–8.57, $p < 0.05$); and intensive outpatient (AOR = 6.40, 95% CI 1.70–24.1, $p < 0.01$) (Scott, 2023). Similarly, maternal/child health nurses trained in MI conducted sessions with postpartum women ($n = 541$, mean age 31.5 years) targeting emotional distress, with MI increasing help-seeking OR = 4.0 (95% CI 1.6–10.1, $p = 0.004$) (Holt, 2017). Veterans with PTSD symptoms ($n = 234$, mean age 50.9 years) who used the PTSD Coach app with clinician support (in-person/telephonic guidance from psychologists) showed OR = 3.91 ($p = 0.016$) for more than two specialty mental health visits among persistent cases (PCL-5 ≥ 33) (Possemato et al., 2023). Here, clinicians addressed motivational, technical or informational barriers.

### Community-based settings (n = 6/7 effective studies)

Effective strategies in community settings included accessible low-intensity support paired with targeted outreach and stigma reduction to remove logistical and psychological barriers. In Nepal, trained female community health facility volunteers conducted proactive universal screening using the Community Informant Detection Tool across 286,838 health facility visits, identifying cases of depression, psychosis, alcohol/substance use and epilepsy and directly referring individuals to primary care (Jordans et al., 2020). This approach significantly increased treatment registration ($r = 0.42$, $p = 0.04$), reflecting a modest but meaningful population-level effect in a low-resource setting.

A 6-week peer-led online support group (HOPE intervention) for adults with moderate-to-severe anxiety ($n = 300$, USA) deployed trained peer leaders in private Facebook groups, facilitating psychoeducation-based discussions (Ugarte et al., 2023). Compared to control groups without peer leaders, intervention participants showed higher help-seeking behavior, such as requesting electronic resources (OR = 10.27, 95% CI 4.52–23.35) and engaging consistently with group posts (OR = 2.8, 95% CI 1.70–4.76). Similarly, telephonic single-session CBT for treatment-seeking (CBT-TS) delivered to military service members and Veterans at risk for suicide ($n = 841$, untreated at baseline) targeted attitudinal and practical barriers through belief modification that achieved significant treatment initiation (OR = 2.53) at

**Table 3.** Summary of included studies

| Author, Year | Country | Population | Sample size and male/female ratio (where reported) | Age | Intervention | Setting | Type of help *(what counts as "help" as reported by authors)* and timepoint of measurement | Delivery agents/ modality | Outcome measures | Reported effectiveness |
|---|---|---|---|---|---|---|---|---|---|---|
| **Age group = 18 years or above** | | | | | | | | | | |
| **Primary outcome: Help-seeking behavior** | | | | | | | | | | |
| Achterbergh et al., 2021 | The Netherlands | Men who have sex with men (MSM) | Randomized, n = 155 Analyzed, n = 150; All males | Median: 43 years IQR (Interquartile Range): 34–51 years | Motivational interviewing + Tailored feedback | STI (sexually transmitted infections) outpatient clinics | Help-seeking from a mental healthcare or addiction treatment service during 1-year follow-up (trimonthly visits T3-T12). | Healthcare providers | Help-seeking behavior was assessed through self-reports and confirmed with records from co-located care or attended clinics, if care was sought outside the STI clinic. | No significant difference between arms |
| Finitsis et al., 2022 | USA | University students with needle anxiety | n = 61 Females: 38 (63.3%) Males: 23 (36.7%) | Mean [SD] = 18.72 + 0.96 years | Single session psychoeducation + Motivational interviewing | University | Treatment or informal support for needle anxiety and related concerns, as listed on the Help-Seeking Behaviors Checklist (e.g., contacting a health professional or service for assistance) since the last assessment (3 months post-intervention). | Undergraduate peer counselors who received MI training and ongoing supervision | Help-seeking behavior was measured using the Help Seeking Behaviors Checklist (3 months post-intervention). | Intervention significantly increased treatment seeking behaviors (IR = 2.41; 95% CI = 1.29, 4.50; $p$ = 0.006) |
| Gilgoff et al., 2023 | USA | Working-age men at risk for suicide | n = 354 All males | Mean [SD] = 39.77 + 11.21 years | Psychoeducation + Self-help resources | Community | Professional help: appointments/assessments with mental health professionals/primary care, professional-led support groups. Non-professional help: peer support groups, online forums/ chats, talking with friends/ family. | Online | Help-seeking behavior was measured using checklists for both professional (e.g., therapy) and non-professional (e.g., peer-based groups) support. Participants indicated whether they sought each form of help, and responses were converted to dichotomous variables (sought/ not sought) over the 3-month intervention period (T2 baseline to T3 12 weeks follow-up). | Intervention significantly increased treatment seeking behaviors (OR = 1.55, $p$ = 0.049), |
| Holt et al., 2017 | Australia | Women in the postnatal period | n = 541 All females | Mean [SD] = 31.5 + 4.7 years | Motivational interviewing | Maternal and child health centers | Help for personal-emotional problems (e.g., depression, anxiety, stress), from any health professional or service, as reported by the participants | Maternal and child health nurses | Help-seeking behavior was assessed at each follow-up by asking women, "Since joining the study, have you sought help for personal-emotional problems (e.g., depression, anxiety, stress)?" over the 12 months post-birth (assessed at 7 weeks, 15 weeks, 12 months). Responses were recorded as "yes" or "no." | Intervention significantly increased treatment-seeking behaviors among emotionally distressed (OR = 4.0, 95% CI: 1.6–10.1, $p$ = 0.004) |
| Ilgen et al., 2022 | USA | Veterans hospitalized following a suicidal crisis | n = 307 Males: 268 (87%) Females: 39 (13%) | Mean [SD] = 47.0 + 13.1 years | Single-session psychoeducation with motivational strategies and behavioral rehearsal | Inpatient | Contact with the Veterans Crisis Line (VCL) for crisis reasons (calls, texts or online chats with the VCL, with or without suicidal intent) | Master's level therapist | Help-seeking behavior was assessed by collecting data on Veterans Crisis Line (VCL) utilization via participant self-report on any contact (e.g., calls, texts, online chats) made for crisis reasons (with or without suicidal intent) during the 12 months post-enrolment. | No significant difference between arms |
| Jordans et al., 2020 | Nepal | Patients visiting health facilities | n = 286,838 Male to female sample sizes not reported | Not reported | Proactive universal screening | Community | Treatment at primary care facilities for depression, psychosis, alcohol use disorder or epilepsy (patients registered in the mental health outpatient register after being identified and referred using the Community Informant Detection Tool) | All-female cadre of community health facility volunteers | Help-seeking behavior was assessed by collecting the total number of patients for whom treatment was recorded during the 6 months after the training of community health volunteers and the start of mental health care plan provision services. | Intervention significantly increased treatment seeking behaviors (U = 36.50, $p$ = 0.04, r = 0.42). |

*(Continued)*

**Table 3.** (*Continued*)

| Author, Year | Country | Population | Sample size and male/female ratio (where reported) | Age | Intervention | Setting | Type of help (what counts as "help" as reported by authors) and timepoint of measurement | Delivery agents/ modality | Outcome measures | Reported effectiveness |
|---|---|---|---|---|---|---|---|---|---|---|
| Kerman et al., 2023 | Canada | Adults experiencing homelessness and unmet mental health needs after hospital discharge. | n = 176 Males/transgender males: 131 (74%) Females: 45 (26%). | Mean [SD] = 44.3 + 12.8 years | Financial incentives | Brief multidisciplinary case management services | Health-related contacts with Coordinated Access to Care for the Homeless (CATCH) service providers | Coordinated Access to Care for the Homeless (CATCH) service providers | Help-seeking behavior was assessed by measuring the number of health-related contacts per month with CATCH service providers over up to 6 months of program enrollment/follow-up | No significant difference between arms |
| King et al., 2023 | USA | University students at risk of suicide | n = 3,363 Males: 1,171 (35.0%); Females: 2,088 (62.2%); Transgender/ genderqueer: 95 (2.8%). | 18 years = 39.7% 19–22 years = 36.6% 23–30 years = 20.1% 31+ years = 3.5% | Screening + Personalized feedback + Referral to treatment | University | Use of mental health services in the past year, including "medication types and counseling or therapy sessions with health professionals." | Mental health professionals included master's or doctoral-level clinicians | Help-seeking behavior was assessed by having participants report their use of mental health services in the past year and within 6 months post-baseline, including medication types and counseling or therapy sessions with health professionals. | No significant difference between arms |
| Lee et al., 2023 | USA | Primary care patients who were at least 18 years old | n = 333,596 Female: 193,583 (58%) Male: 140,013 (42%) | Mean [SD] = 49.75 + 18.08 years | Health system level support | Primary care | Specialist treatment for alcohol use disorder within the health system, including "new AUD diagnoses on the day of a primary care visit" and subsequent "initiation and engagement in in-person treatment" for AUD. | Primary care staff | Help-seeking behavior was assessed by identifying new AUD diagnoses on the day of a primary care visit (with no prior AUD diagnosis in the past year) and subsequent initiation and engagement in in-person treatment (initiation within 14 days of new diagnosis +2 more visits within 30 days). Rates measured monthly during intervention vs. usual care periods | No significant difference between arms. |
| Possemato et al., 2023 | USA | Veterans with PTSD symptoms | n = 234 Males: 211 (90.2%) Females: 23 (9.8%). | Mean [SD] = 50.91 ± 5.43) years | Self-help mobile application for PTSD survivors (PTSD Coach app) with support and guidance on use by a clinician during in-person and/or telephonic sessions | Primary care | VA mental health visits (PCMHI, CS PTSD Coach sessions, specialty care psychotherapy/ psychiatry) during intervention (weeks 0–8) | Psychologists and masters-level clinicians | Help-seeking behavior was measured through hospital administrative data showing 2 mental health visits for PTSD completed. Engagement measured weeks 9–24 post-treatment; intervention period weeks 0–8. | Intervention significantly increased treatment seeking behaviors OR = 3.91 (95% CI; $p = 0.016$) |
| Ries et al., 2022 | USA | Adult patients with an SUD | n = 616 Males: 540 (59.6%) Females: 350 (38.6%) Trans/Non-conforming: 4 (0.4%) Missing: 12 (1.3%) | Mean [SD] = 37.5 ± 12.0 years | Single session psychoeducational module | Outpatient program | Help-seeking behaviors for suicidal thoughts/feelings (self or others, e.g., calling crisis/ suicide hotline, friends/family) via PARS Help-Seeking Scale. | Counselors trained in the intervention | Help-seeking behavior was measured by the Preventing Addiction Related Suicide (PARS) Help-Seeking Scale over past month (assessed at baseline, post-treatment, 1/3/6 months). | Intervention significantly increased treatment seeking behaviors ($d = 0.16$; 95% CI: 0.01 to 0.32; $P = 0.04$). |
| Scott et al., 2023 | USA | Primary care patients who were referred to SUD treatment after identification | n = 266 Males: 173 (65%) Females: 93 (35%) | Mean age = 48.3 years Range: 19–53 years | Screening, brief intervention and referral to treatment (SBIRT) + MI-based case management | Primary care | Self-reported SUD treatment (residential, intensive outpatient, outpatient, medication-assisted, other) in past 90 days (3 months post-baseline). | Linkage worker | Help-seeking behavior was measured by asking participants during the 3-month follow-up interview about the number of days they received specific types of SUD treatment. | Intervention significantly increased treatment seeking behaviors (AOR = 4.59; $p = <0.001$), specifically for residential treatment (AOR =3.31; $p = <0.05^*$) and intensive outpatient treatment (AOR = 6.40; $p = <0.01$). |

(*Continued*)

**Table 3.** (Continued)

| Author, Year | Country | Population | Sample size and male/female ratio (where reported) | Age | Intervention | Setting | Type of help (what counts as "help" as reported by authors) and timepoint of measurement | Delivery agents/ modality | Outcome measures | Reported effectiveness |
|---|---|---|---|---|---|---|---|---|---|---|
| Shafran et al., 2019 | UK | Adults with significant levels of anxiety and depression | Randomized, n = 6,725; Analyzed, n = 306 Females: 217 (70.9%) Males: 89 (29.1%) | Mean [SD] = 27.15 ± 9.72 years | Psychoeducation + Self-help resources + Symptom monitoring | Community | Receipt of any mental health treatment (GP, IAPT, private therapist; psychological therapy, medication, both, other) since study start (over 30 days). | Online | Help-seeking behavior was assessed by reporting receipt of treatment within the 30 days since starting the study. | No significant differences between the arms |
| Stecker et al., 2023 | USA | At-risk military members and Veterans | n = 841 Females: 62 (7.4%) Males: 714 (84.9%) | Mean [SD] = 31.7 ± 7 years Range: 21 to 67 years | Single session cognitive behavioral therapy for treatment-seeking (CBT-TS) | Community | Behavioral health treatment initiation (attending ≥1 mental health appointment for suicidality/related conditions; VA or non-VA, physician / psychologist/ other). | PhD or masters-level clinicians/ Telephonic | Help-seeking behavior was measured by assessing treatment utilization: (a) Treatment initiation was determined by asking participants if they had attended a mental health appointment. (b) Treatment sessions attended were assessed by querying participants about the number of mental health treatment sessions they had attended. | Intervention significantly increased treatment seeking behaviors (initiation) (OR = 1.31, 95% CI 1.01–1.86). |
| Ugarte et al., 2023 | USA | 18 years or older adults with anxiety, currently not on anxiety medication | n = 300 Females: 244 (81%) Males: 56 (19%) | Mean [SD] = 39.14 ± 15.59 years | Peer-led online support group including communication led by peer leaders on psychoeducation-based topics | Community | Requests for electronic resources (e-resources) on anxiety reduction (e.g., cognitive-behavioral therapy info), measured over 6 weeks | Peer leaders/ Online | Help-seeking behavior was measured by recording participants' requests for e-resources and online engagement, including actions like commenting, posting, or reacting to posts. Online engagement was measured biweekly by tracking consistent interaction with posts in their Facebook group. | Intervention significantly increased treatment seeking behaviors [request of e-resources (OR = 10.27, 95% CI 4.52–23.35) and online engagement (OR = 2.8, 95% CI 1.70–4.76)] |
| **Primary Outcome: Help-seeking Intention** | | | | | | | | | | |
| Amsalem, Lazarov et al., 2022 | USA | Health workers | n = 350 Females: 260 (74%) Males: 90 (26%) | Mean [SD] = 34.8 ± 11.5 years Range: 18–70 years | Social contact-based video interventions. | Community | Intentions for psychological counseling/professional help for emotional problems, e.g., wanting counseling if upset long-term, believing professional help solves problems better than alone), assessed over 30 days (pre/post-intervention, 14/30-day follow-ups). | Online | Help-seeking intention was measured using three items from the Attitudes Toward Seeking Professional Psychological Help Scale (ATSPPH-SF) assessing openness to help-seeking. | Intervention significantly increased treatment-seeking intentions [95% CI 9.3–10.1], P < 0.001, Cohen's d = 0.74 |
| Amsalem, Wall et al., 2023 | USA | Healthcare workers | n = 1,402 Females: 1152 (82%) Males: 250 (18%) | Mean [SD] = 28.9 ± 9.1 years Range: 18–66 years | Social contact-based video interventions | Community | Intentions for psychological counseling/professional help for emotional problems, e.g., wanting counseling if upset long-term, professional help better than alone; treatment-related stigma (SSOSH–3), assessed over 30 days (baseline/post-intervention, 14/30-day follow-ups). | Online | Help-seeking intention was assessed using three items from the Attitudes Toward Seeking Professional Psychological Help Scale–Short Form (ATSPPHSF), measuring openness to treatment seeking. | Intervention significantly increased treatment-seeking intentions (p < 0.001, effect size [ES] = 21%), |

(Continued)

**Table 3.** (*Continued*)

| Author, Year | Country | Population | Sample size and male/female ratio (where reported) | Age | Intervention | Setting | Type of help (what counts as "help" as reported by authors) and timepoint of measurement | Delivery agents/ modality | Outcome measures | Reported effectiveness |
|---|---|---|---|---|---|---|---|---|---|---|
| Fernandez et al., 2022 | Australia | General population | n = 271 Females: 162 (60%) Males: 91 (34%) Other: 18 (7%) | Mean [SD] = 31.65 ± 11.51 years Range: 18–83 years | Media-based intervention, including a short educational-style video, pseudo-research highlights and a pseudo-magazine article presenting different formats of information about a disorder | Community | Intentions to seek professional help if experiencing problems like vignette (schizophrenia symptoms); intentions to provide help if knowing someone like vignette; prosocial support behavior (interest in volunteering as an email pen-pal for people with schizophrenia), measured pre/post-intervention | Online | Help-seeking intention was assessed with a single-item question ":If you were experiencing a problem like James', how likely would you be to seek help?" Responses were rated on a 5-point scale (1 = extremely unlikely, 5 = extremely likely). | No significant difference between the arms. |
| Hollar and Siegel, 2020 | USA | People with heightened depressive symptomatology | Study 1, n = 61 Females: 44.3% Males: 55.7% Study 2, n = 297 Females: 50.8% Males: 48.5% Study 3, n = 574 Females: 50.3% Males: 49.0% | Mean [SD] Study 1: 36.16 ± 11.32 years Study 2: 34.7 ± 11.56 years Study 3: 34.67 ± 10.51 years | Writing task to induce self-distancing mental state | Community | Intentions to seek help from a range of formal and informal sources for personal or emotional problems (as listed on the General Help-Seeking Questionnaire) measured post-intervention | Online | Help-seeking intention was measured using the General help-seeking questionnaire (GHSQ) | Intervention significantly increased treatment-seeking intentions (M = 3.84, SE = 0.20, p = 0.011, $\eta^2$p = 0.11, 95% CI [0.19–1.41]). |
| King et al., 2023 | Australia | Workers in the construction industry | n = 1,084 Females: 71 (6.6%) Males: 940 (87.0%). | Mean age = 33.50 years. | Psychoeducation + Dissemination of resources + Social contact videos | Community | Intentions to seek help for emotional problems/suicidal thoughts from formal sources (mental health professionals, doctors/GPs, phone helplines), informal sources (partners, relatives, friends, religious leaders, work supervisors), MATES workers/Connectors, workmates, or no one (via modified GHSQ), measured at 3-month follow-up | Program workers (field officers employed by the suicide prevention program to provide ongoing support to sites) and Connectors (volunteer "gatekeepers" who help at-risk workers access help and support) | Help-seeking intention was measured using the General help-seeking questionnaire (GHSQ) | No significant differences between the arms. |
| King et al., 2018 | Australia | Men aged 18 and above | Randomized, n = 354 Analyzed, n = 337 All males | Mean [SD] = 38.8 ± 13.56 years | Social contact-based documentary | Community | Intentions to seek help for personal/emotional problems from intimate partner, friend, doctor, or other sources including online health chat rooms, online searches, social media (via GHSQ); intentions to recommend male/female friend seek help from same sources (modified GHSQ), measured 4 weeks post-viewing | Online | Help-seeking intention was measured using the General help-seeking questionnaire (GHSQ) | Intervention significantly increased treatment-seeking intentions (Linear Regression Coef. 2.06, 95% CI 0.48 to 3.63, P = 0.01) |
| Kruger et al., 2022 | USA | Research participants from undergraduate psychology courses | n = 163 Females: 121 (74.2%) Males: 42 (25.8%) | Mean [SD] = 21.05 ± 2.20 Range: 19–29 years | Social contact-based video intervention | University | Intentions to seek counseling/ therapy for mental health problems (e.g., depression; via ISCI, 17 items), measured pre/post-intervention | Online | Help-seeking intention was measured using Intentions to Seek Counseling Inventory | Intervention significantly increased treatment-seeking intentions (p < 0.01, partial $\eta^2$p = 0.13). |

(*Continued*)

**Table 3.** (*Continued*)

| Author, Year | Country | Population | Sample size and male/female ratio (where reported) | Age | Intervention | Setting | Type of help (*what counts as "help" as reported by authors*) and timepoint of measurement | Delivery agents/ modality | Outcome measures | Reported effectiveness |
|---|---|---|---|---|---|---|---|---|---|---|
| Mason et al., 2022 | New Zealand | Undergraduate students | *n* = 133 Females: 121 (91.0%) Males: 11 (8.3%) | Mean [SD] = 20.17± 2.31 years Range: 18–35 years | Promotion of help-seeking resources in different formats (video vs. infographic) | University | Intentions to seek help from university health service (counseling /psychological services) or other professionals (doctor/GP, mental health professional, religious advisor) if problem caused distress/ interfered with work; measured at baseline and follow-up at 1 year (Study 1) or across 6 weeks (Study 2) | Online | Help-seeking intention was assessed using a five-point Likert scale. Participants responded to questions about their likelihood of seeking help from the university's health service or other sources for distressing problems. | Intervention (Infographic) significantly increased treatment-seeking intentions (M = 3.52, SD = 1.02 vs. M = 3.05, SD = 1.03, *p* = 0.035).,. |
| Smith et al., 2023 | Australia | Adults aged 18 and above | *n* = 676 Females: 356 (52.7%) Males: 319 (47.2%). | Mean [SD] = 42.33 ± 13.91 years | Alternative terminology for diagnostic labels (e.g., "depression," "burnout," "functional impairment syndrome" vs. no label) and mental health providers (clinical psychologist vs. mind coach) | Community | Intentions to seek help from clinical psychologist or mind coach measured immediately post-intervention | Online | Help-seeking intention was assessed by having participants rate their likelihood of seeking help from a clinical psychologist or mind coach on a 5-point scale, ranging from "very unlikely" to "likely." | No significant difference between the arms for diagnostic labels Intervention significantly increased treatment-seeking intentions (MD = 0.43, 95% CI: 0.25–0.60) |
| Straszewski and Siegel, 2018 | USA | People with depressive symptomatology | Study 1, *n* = 135 Females: 60.74% Males: 39.26% Study 2, *n* = 136 Females: 54.41% Males: 45.59% | Mean [SD] = 34.90 ± 11.86 years | Writing task with prompts to induce positive emotions | Community | Intentions to seek help from family member, mental health professional, primary care physician, friend, romantic partner, parent, website, or at least one person measured immediately post-intervention | Online | Help-seeking intention was measured using the General help-seeking questionnaire (GHSQ) | Intervention significantly increased treatment-seeking intentions (M = 4.09, SE = 0.15, *p* = 0.032, partial $\eta^2 p$ = 0.04.) |
| Till et al., 2024 | Austria | German-speaking individuals of the general population aged 18 years or older | *n* = 334 Females: 255 (76.3%) Males: 77 (23.1%) | Mean [SD] = 29.1 ± 10.65 years | Media-based intervention: educative news articles featuring an expert discussing suicide prevention presented with varying narratives. (e.g., one article highlighted prevalence of suicide, another highlighted professional help resources, and the other emphasized on how everyone can help to prevent suicide) | Community | Intentions to seek help for personal/emotional problems or suicidal thoughts from personal contacts (e.g., family/ friends), professional help (e.g., mental health professionals/ helplines) or online resources measured immediately post-intervention | Online | Help-seeking intention was measured using the General help-seeking questionnaire (GHSQ) | Intervention significantly increased treatment-seeking intentions (*p* = 0.038, $\eta^2 p$ = 0.019) |
| **Primary Outcome: Help-seeking Attitude** | | | | | | | | | | |
| Aisenberg-Shafran and Shturm, 2022 | Israel | Elderly persons with normative functioning | *n* = 24 All females | Mean [SD] = 73.13 ± 5.67 years | The Mindfulness-Based Intervention for Seniors (MBIS) that included practices and techniques to enhance mindfulness, reduce stress and improve overall well-being. | Geriatric nursing home | Attitudes toward seeking mental health services assessing psychological openness/help-seeking propensity/indifference to stigma measured before/after 8 weekly 30-min group sessions | Instructor with 2 years of experience | Help-seeking attitude was measured using the Inventory of Attitudes Toward Seeking Mental Health Services (IASMHS) | Intervention significantly increased treatment-seeking attitudes (*p* < 0.05, $\eta^2 p$ = 0.3) |

(*Continued*)

**Table 3.** (*Continued*)

| Author, Year | Country | Population | Sample size and male/female ratio (where reported) | Age | Intervention | Setting | Type of help *(what counts as "help" as reported by authors)* and timepoint of measurement | Delivery agents/ modality | Outcome measures | Reported effectiveness |
|---|---|---|---|---|---|---|---|---|---|---|
| Amsalem, Wall et al., 2022 | USA | Non-healthcare essential workers | *n* = 2,734 Females: 1702 (62%) Males: 1032 (38%). | Mean [SD] = 27.1 ± 9.3 years Range: 18–73 | Social contact-based video interventions. | Community | Openness to seeking professional psychological help if worried/upset long period, future counseling, professional help for emotional problems; self-stigma of seeking therapy measured at baseline/post-intervention/14-day/30-day follow-up | Online | Help-seeking attitude was assessed using the three "openness to help-seeking" items from the Attitude Toward Seeking Professional Psychological Help (ATSPPH) scale – Short Form. | Intervention significantly increased treatment-seeking attitudes (*P* < 0.0001; *d* = 0.18) |
| Conceição et al., 2022 | Portugal | First-year University students | Randomized, *n* = 969; Analyzed, *n* = 626 Females: 380 (60.7%) Males: 246 (39.3%). | Mean [SD] = 19.85 + 1.48 years | Psychoeducation + social contact | University | Attitudes toward seeking professional psychological help measured pre-intervention/post-intervention/5-month follow-up | Online | Help-seeking attitude was measured using The Attitudes Toward Seeking Professional Psychological Help (ATSPPH) | Intervention significantly increased treatment-seeking attitudes (*p* < 0.001, R² = 0.36; Cohen's f² = 0.04) |
| Milner et al., 2018 | Australia | Male construction workers | Randomized, *n* = 682; Analyzed, *n* = 478 All males | Mean age = 49.61 years | MHL + promotion of help-seeking resources + encouraging social connection with others | Community | Professional help for depression and related mental health problems, as reflected in the "help-seeking" subscale of the Self-Stigma of Depression Scale (e.g., willingness or embarrassment about seeking professional help and attitudes toward antidepressant use), measured at baseline/6-week post-intervention | Online | Help-seeking attitude was assessed using the help-seeking subscale of the Self-Stigma of Depression Scale (SSDS). This subscale consists of four items measuring attitudes such as feeling embarrassed about seeking professional help for depression and perceptions about antidepressant use (mean 9.89, S.D. = 2.75). | No significant differences between the arms |
| **Primary Outcomes: Help-seeking Behavior, Intention and Attitude** | | | | | | | | | | |
| Martin et al., 2020 | USA | College athletes | *n* = 107 Females: 58 (54.2%) Males: 49 (45.8%) | Mean [SD] = 19.70 ± 2.10 years Range: 18–29 years | MHL + Stigma reduction strategies + Promotion of help-seeking resources + social-contact + Open discussions | University | Attitudes/intentions toward professional psychological help for eating pathology and general mental health measured at baseline /post-intervention/ 6-week follow-up Help-seeking behavior (any source, including mental health professionals/friends/ family/physicians/athletic trainers) for mental health concerns (self or referral) over 6-week period post-intervention | Not mentioned | Attitude was assessed using the Attitudes Toward Seeking Professional Psychological Help-Short Form (ATSPPH-S). Intention was measured using the General Help Seeking Questionnaire (GHSQ). Behavior was measured using investigator-created questions, including whether participants sought help, where they sought help, reasons for seeking help, and if they referred someone for help. Help-seeking behavior was tracked retrospectively over six weeks post-intervention via self-report (e.g., consultation completion, seeking therapy). | Intervention significantly increased help-seeking attitudes: (*p* < .001, η² = 0.11 Intervention significantly increased help seeking intentions: (*p* = 0.003, η² = 0.09) Intervention significantly increased help seeking behavior: (OR 5.06, *p* = 0.01) |
| Tobias et al., 2022 | USA | Adults with social anxiety | Randomized, *n* = 267; Analyzed, *n* = 241 Females: 167 (69.29%) Males: 70 (29.05%) Transgender/ non-binary: 4 (1.66%) | Mean [SD] = 34.44 ± 10.38 years | A brief, single-session online intervention based on psychoeducation on Social Anxiety Disorder (SAD) symptoms + Cognitive Behavioral Therapy (CBT) for SAD, + | Community | Treatment-seeking behaviors for social anxiety (10-step checklist: preparatory steps, e.g., researching treatments, contacting providers, scheduling /attending appointments for in-person therapy/ bibliotherapy/online therapy/alternative treatments), measured over 1-month post-intervention | Online | Attitude was assessed using seven semantic differential scale items (e.g., "harmful"/ "beneficial"), yielding a total score with high internal consistency (Cronbach's α ranged from .88 to .91). Intention was measured through 13 Likert scale items covering various treatment modalities, resulting in a total | Intervention significantly increased treatment-seeking attitudes (*d* = 0.33, 95% CI, *p* < .001)] Intervention significantly increased help-seeking intentions (*d* = 0.18, 95% CI, *p* = 0.06) |

**Table 3.** (*Continued*)

| Author, Year | Country | Population | Sample size and male/female ratio (where reported) | Age | Intervention | Setting | Type of help (what counts as "help" as reported by authors) and timepoint of measurement | Delivery agents/ modality | Outcome measures | Reported effectiveness |
|---|---|---|---|---|---|---|---|---|---|---|
| | | | | | brief motivation intervention. | | (self-report at 1-month follow-up). | | score with high internal consistency (Cronbach's α ranged from .94 to .96). Behavior was assessed via a checklist of 10 sequential steps capturing total steps taken and sums within treatment categories. | Intervention significantly increased help-seeking behaviors ($p$ = .04, $\eta^2$ = .02 CI90% = [.00 .07]) |
| **Primary Outcomes: Help-seeking Behavior and Intention** | | | | | | | | | | |
| Coleman et al., 2019 | USA | Undergraduates | Randomized, $n$ = 117; Analyzed, $n$ = 69 Females: 51 (74%) Males: 18 (26%) | 20.5 Mean [SD] = 20.5 ± 7.3 years | Online, interactive suicide prevention gatekeeper training | University | Use of counseling services at the campus counseling center (students who completed the gatekeeper training were compared with all other college students in terms of actual help-seeking from the counseling center) during the academic year following training (e.g., after training date until end of academic year, approx. 8 months) | Online | Intention was assessed using single items that measured the likelihood of personal help-seeking. Behavior was measured by analyzing administrative data from the training center database, specifically focusing on students who completed the gatekeeper program. The help-seeking rate among these students was contrasted with that of all other college students during the same period. | Intervention significantly increased help-seeking intentions ($d$ = 0 .64, $p$ < .05) Intervention significantly increased help-seeking behavior (OR = 2.3, $p$ < .001). |
| Nickerson et al., 2020 | Australia | Male refugees from Arabic, Farsi or Tamil speaking backgrounds | $n$ = 103 All males | Mean [SD] = 39.37 ± 9.88 years | Psychoeducation + Social contact + promotion of help-seeking resources | Community | Help-seeking behaviors from 11 sources (friend, spouse/ partner, parent, son/daughter, other family member, caseworker, community leader, doctor/general practitioner, teacher, religious leader, mental health professional), measured as number of different types of help-seeking sources accessed in the past 2 weeks, measured over post-intervention (4 weeks) and 1-month follow-up (8 weeks). | Online | Intention was measured using an 11-item adapted version of the General Help-Seeking Questionnaire (GHSQ). Behavior was measured using an 11-item adapted version of the Actual Help-Seeking Questionnaire. | Intervention significantly increased help-seeking behaviors ($B$ = 0.69, 95% CI 0.19–1.18, $p$ = 0.007) Intervention significantly increased Help-seeking intentions ($d$ = 0.27, $p$ = 0.027) |
| **Primary Outcomes: Help-seeking Intention and Attitude** | | | | | | | | | | |
| Han et al., 2023 | China | Sexual and Gender Minority (SGM) young adults | $n$ = 142 Females: 39 (27.5%) Males: 33 (23.2%) Transgender/ non-binary/ others: 70 (49.3%) | Mean [SD] = 22.17 ± 2.80 years | Psychoeducation + facilitator-led group discussions around mental health + promotion of help-seeking resources such as brochures. | Community | General help-seeking intentions for emotional problems/suicidal ideation from friends/family/partner/ workmate/parent/child/ relative/doctor/teacher/ religious leader /counselor/ psychiatrist/psychologist/ mental health hotline) and attitudes toward professional psychological help measured at post-discussion/1-month/3-month follow-ups | Group facilitators had a master's degree in psychology/ Online | Intention was measured using General Help-Seeking Questionnaire (GHSQ) Attitude was measured using Attitudes Toward Seeking Professional Psychological Help Scale Short Form (ATSPPH-SF) | Intervention significantly increased help-seeking intentions for suicidal ideation (Mean difference = 0.19, 95% CI [0.06, 0.33], $p$ = 0.018) Intervention significantly increased help-seeking intentions for emotional issues (Mean difference = 0.17, 95% CI [0.05, 0.28], $p$ = 0.013) No significant difference between arms for help-seeking attitudes. |

(*Continued*)

| Author, Year | Country | Population | Sample size and male/female ratio (where reported) | Age | Intervention | Setting | Type of help (*what counts as "help" as reported by authors*) and timepoint of measurement | Delivery agents/ modality | Outcome measures | Reported effectiveness |
|---|---|---|---|---|---|---|---|---|---|---|
| Hollar and Siegel, 2023 | USA | Adults with elevated depressive symptomatology who had not yet sought help for current feelings of depression | n = 870 Females: 62.8% | Mean [SD] = 35.91 ± 11.52 years | Writing-based task to induce the mental state of self-distancing, where participants imagined seeking help for depression from the perspective of their ideal future self. | Community | Help-seeking intentions from 8 sources, e.g., romantic partner/ close friend/parent/other family/counselor or psychologist/psychiatrist/ doctor or GP/at least one source; attitudes toward help-seeking (5 semantic differential items); self-stigma of seeking professional help assessed post-intervention | Online | Attitude toward seeking help for depression was assessed using a 7-point semantic differential scale (α = .93), based on Osgood's traditional measures (Osgood et al., 1957). Intention was measured using the GHSQ. | Intervention significantly increased help-seeking attitudes ($p$ = .005, ηp² = 0.013) Intervention significantly increased Help-seeking intentions l ($p$ = .02, ηp² = .01) |
| Hussain and Alhabash, 2022 | USA | Students living with depression | Randomized, n = 366; Analyzed, n = 50 Females: 169 (46.2%) Males: 196 (53.6%) | Mean [SD] = 21.63 ± 1.98 years | Public service announcement (PSA) videos to evoke nostalgia in participants | University | Contacting the campus counseling center for personal or emotional issues in the future (behavioral intention to contact the campus counseling center, and attitudes toward the counseling center) assessed post-intervention | Online | Attitude toward the counseling center was assessed using a seven-point semantic differential scale measuring dimensions like Unhelpful-Helpful and Unpleasant-Pleasant (α = .89). Behavioral intention to contact the campus counseling center was measured with three study-specific items (α = .91), rated on a seven-point Likert scale indicating intent to seek help for personal or emotional issues in the future. | Intervention significantly increased help-seeking attitudes (M. = 5.25, SD =1.20, $p$ = 0.056, h²p = 0.03). No significant differences between arms for help-seeking intentions |
| Hussain and Alhabash, 2020 | USA | Adults experiencing mild to severe levels of depression | n = 148 Females: 88 (59%) Males: 60 (41%) | Mean [SD] = 39.64 ± 11.76 years | A video public service announcement (PSA) intervention to evoke nostalgia in participants, with the nostalgic content manipulated to be either positive, negative, or a mix of both positive and negative elements. | Community | Help from a counselor when feeling depressed, as reflected in participants' willingness to seek help from a counselor in the future when feeling depressed; perceived behavioral control over seeking help from counselor; descriptive/injunctive subjective norms re seeking help from counselor when depressed assessed post-intervention | Online | Attitude toward help-seeking from a counselor when feeling depressed was measured using four semantic-differential items: Bad-Good, Dislike-Like, Unpleasant-Pleasant and Not Beneficial-Beneficial. Intention to seek help was measured with four items on a 7-point Likert scale ranging from Strongly Disagree to Strongly Agree, assessing willingness to seek help from a counselor in the future when feeling depressed. | No significant difference between arms |
| Seidman et al., 2022 | USA | University students | n = 172 Females: 118 (68.6%) Males: 51 (29.7%) | Mean [SD] = 19.19 + 2.41 years Range: 18–43 years | Two interventions: (a) Group counseling; (b) Group counseling combined with self-affirmation task | University | Attitudes toward seeking help from mental health professional; behavioral intentions to seek help from mental health professional; public stigma toward receiving psychological help; self-stigma of seeking psychological help measured pre/post session | Doctoral students in Counseling Psychology | Attitude was measured by using the Mental Help Seeking Attitudes Scale Intention was measured by using the Mental Help Seeking Intentions Scale. | Intervention significantly increased help-seeking attitudes γ00 = 0.50, padj = .002, 95% CI = [0.20, 0.81] No significant differences between arms for help-seeking intentions |
| Tay, 2022 | Singapore | University students | n = 174 Females: 124 (71.3%) Males: 50 (28.7%) | Mean age = 20.47 years. | Online MHL sessions + promotion of help-seeking resources | University | Intentions to seek professional help if experiencing vignette character's problems (single item); attitudes relating to barriers to help-seeking (e.g., cost, stigma, side effects), interventions/help sources/ medications for depression (e.g., antidepressants, counseling, self-help) assessed pre-post intervention. | Online | Intention was assessed with a single question asking participants if they would seek professional help for problems like those experienced by the vignette character. Attitudes was assessed through questions about barriers to help-seeking, perceptions of interventions, sources of help, and medications for mental health conditions. | No significant differences between the arms for help-seeking intentions Intervention significantly increased help-seeking attitudes: Greater acknowledgement of antidepressants (U = 2,774.50, $p$ < 0.001) and antipsychotics (U = 2,799, $p$ = 0.003) |

**Table 3.** (*Continued*)

| Author, Year | Country | Population | Sample size and male/female ratio (where reported) | Age | Intervention | Setting | Type of help (what counts as "help" as reported by authors) and timepoint of measurement | Delivery agents/ modality | Outcome measures | Reported effectiveness |
|---|---|---|---|---|---|---|---|---|---|---|
| **Age group = 18 years or below** | | | | | | | | | | |
| **Primary Outcome: Help-seeking Behavior** | | | | | | | | | | |
| Casañas et al., 2022 | Spain | School students (adolescents) | n = 1,032 Females: 512 (49.6%) Males: 520 (50.4%) | Mean [SD] = 14.2 + 0.58 years | MHL session + Social contact | School | Past or current psychological/ medication treatment for MH problem (lifetime); intended help-seeking from friend / parent /teacher/MHP/no one if experiencing MH problem assessed during baseline/ 6 months/ 12 months | Mental health nurses | Help-seeking behavior was assessed using: (a) If participants received psychological or medicinal treatment for mental health issues at any point in their lifetime. (b) The first item from the General Help-Seeking Questionnaire (GHSQ) | No significant difference between arms |
| Grupp-Phelan et al., 2019 | USA | Adolescents at non-psychiatric Emergency Department (ED) visits | Randomized, n = 168; Analyzed, n = 159 Females: 126 (79.2%) Males: 33 (20.8%) | Mean [SD] = 15.0 + 1.5 years | Motivational interviewing | Pediatric emergency departments | Initiation of outpatient mental health treatment (≥1 visit; verified by agency/parent report); number of MH appointments attended; assessed at 2 months and 6 months post-ED discharge | Social workers trained in the intervention | Help-seeking behavior was assessed using hospital administrative data indicating completion of two mental health visits for PTSD. | No significant difference between arms |
| Link et al., 2020 | USA | School students (adolescents) | n = 416 Females: 233 (56%) Males: 183 (44%) | Mean = 11.5 years | Anti-stigma and MHL curriculum + Group discussions + Homework exercises | School | Formal treatment for a mental health problem, defined as "taking medicine for a mental health problem" and/or "talking to a therapist or counselor about a mental health problem" assessed at 6/12/18/24 months post-intervention. | Teachers | Help-seeking behavior was assessed by asking whether youth had taken medicine for a mental health problem or talked to a therapist or counselor about a mental health problem (coded "1" if either or both, coded "0" if neither). | Intervention significantly increased treatment seeking behaviors among those with high symptom levels (OR = 3.90, CI = 1.09–13.87) |
| Lubman et al., 2020 | Australia | School students (adolescents) | n = 2,447 Females: 1231 (50.3%) Males: 1216 (49.7%) | Mean [SD] = 14.9 ± 0.5 years Range: 14–15 years | MHL + Gatekeeping skills training | School | Help-seeking (ever) for stress/ anxiety, depression or AOD problems from formal (health professionals: doctor/ counselor/nurse/psychologist) or informal (parents/ friends/ siblings/other family/teachers) sources, timeframe within which sought (past 6 weeks/ past 6 months/ past 12 months/>12 months) assessed at 6wk/6mo/ 12 months post-baseline | Not reported | Help-seeking behavior was assessed using the Actual Help Seeking Questionnaire (AHSQ), | No significant difference between arms |
| Lustig et al., 2023 | Austria, Estonia, Germany, France, Hungary, Ireland, Israel, Italy, Romania, Slovenia and Spain | School students (adolescents) | n = 4,172 Male/female sample not reported | Mean [SD] = 15 ± 0.9 years | Screening + Referral to treatment | School | Use of professional MH services (medication/ one-to-one therapy/ group therapy/advice from health professional) from baseline (within 12 months) assessed at 12 months post-baseline. | Psychologist or psychiatrist | Help-seeking behavior was assessed based on whether participants received help from a health professional (yes) or sought help within the lay support system or did not seek any help (no). In this study, 'service use' specifically refers to the use of specialist mental health services (psychologists, psychiatrists, psychiatric clinics). | No significant difference between arms |

(*Continued*)

**Table 3.** (Continued)

| Author, Year | Country | Population | Sample size and male/female ratio (where reported) | Age | Intervention | Setting | Type of help (what counts as "help" as reported by authors) and timepoint of measurement | Delivery agents/ modality | Outcome measures | Reported effectiveness |
|---|---|---|---|---|---|---|---|---|---|---|
| Mori et al., 2022 | Japan | School students (adolescents) | n = 116 Females: 61 (52.6%) Males: 55 (47.4%) | Range: 12–13 years | MHL sessions + Social contact | School | Help-seeking behaviors (talked to family/friend about MH; visited MH website; consulted someone about MH) in past 3 months assessed post-program and 3 months post-baseline. | Health/physical education teachers | Help-seeking behavior was assessed with a four-question self-administered questionnaire, asking participants about discussing mental illness with family, visiting mental health websites, talking to friends and consulting someone about mental illness in the past three months. Participants responded with "yes," "no," or "don't know," with "yes" indicating engagement in help-seeking behavior. | No significant difference between arms |
| Parikh et al., 2021 | India | School students (adolescents) | n = 3,587 Females: 1,551 (43.2%) Males: 2,036 (56.8%) | Mean [SD] = 15.8 ± 0.06 years | Single classroom MHL sessions | School | Referrals to school counseling for common MH problems (self/ teacher/drop-box); assessed over the 12 week trial period | Lay counselor | Help-seeking behavior was assessed by calculating the "referral proportion," which represents the proportion of adolescents in the participating classes referred for counseling. | Intervention significantly increased treatment seeking behaviors (OR = 111.36, 95% CI 35.56 to 348.77, p < 0.001). |
| **Primary Outcome: Help-seeking Intention** | | | | | | | | | | |
| Amsalem et al., 2021 | USA | English-speaking youth | n = 1,183 Females: 556 (47.0%) Males: 627 (53.0%) | Mean [SD] = 16.8 ± 1.2 years Range: 14–18 years | Social contact-based video interventions. | Community | Treatment-seeking intentions (likelihood of seeking help from friend/partner/parent/ relative/mental health professional/helpline/doctor/ GP/religious leader/ other/not seek for emotional/personal problems or suicidal thoughts) assessed immediately post-intervention | Online | Help-seeking intention was measured using the General Help-Seeking Questionnaire (GHSQ; p < .001) | Intervention significantly increased treatment-seeking intentions (d = ranged from 0.10 to 0.25.) Composite score not reported |
| O'Dea et al., 2021 | Australia | Students | n = 1854 Females: 930 (51.6%) Males: 872 (48.4%) | Mean [SD] = 14.3 ± 0.87 years | Promotion of help-seeking resources + MHL sessions | School | Intentions to seek help (from 13 sources: friend/partner/ parent/relative/family friend/ teacher/adult/school counselor/ GP/ mental health professional/ helpline/ religious leader/not seek/ other) for general mental health problems assessed at 12 weeks post-baseline | Online | Help-seeking intention was measured using the General help-seeking questionnaire (GHSQ) | Intervention significantly increased treatment-seeking intentions; Cohen's d = 0.10, 95%CI: −0.02–0.21 |
| **Primary Outcomes: Help-seeking Behavior and Intention** | | | | | | | | | | |
| Calear et al., 2022 | Australia | Adolescents | n = 1,633 Females: 58.2% | Mean [SD] = 13.46 ± 1.23 years Range: 11–17 years | Universal school-based peer leadership program | School | Help-seeking for MH problem past 3 months (yes/no; from formal: psychologist/other mental health professional/ GP/counselor; informal: parent/other family/teacher/ other trusted adult) assessed pre/post/6 months/18 months. Intentions to seek help (likelihood from the same 8 formal/informal adult sources) assessed pre/post/6 months/ 18 months | Online | Intention was measured using the General Help-Seeking Questionnaire (GHSQ). Behavior was assessed using a single yes/no item to determine if participants had sought help for a mental health issue in the past 3 months. | No significant difference between arms |

(Continued)

**Table 3.** (*Continued*)

| Author, Year | Country | Population | Sample size and male/female ratio (where reported) | Age | Intervention | Setting | Type of help (*what counts as "help" as reported by authors*) and timepoint of measurement | Delivery agents/ modality | Outcome measures | Reported effectiveness |
|---|---|---|---|---|---|---|---|---|---|---|
| **Age group = Mixed (adults + children/ adolescents)** | | | | | | | | | | |
| **Primary Outcome: Help-seeking Behavior** | | | | | | | | | | |
| Reinauer et al., 2021 | Germany | Youths with chronic medical conditions and comorbid anxiety or depression | *n* = 164 Females: 97 (59%) Males: 67 (41%) | Mean [SD] = 15.2 ± 1.9 years Range: 12–20 years | Motivational interviewing | Outpatient clinics | Utilization of mental health services, defined as "at least one contact with a mental health service (psychiatrist, psychotherapist, counseling center, inpatient treatment)" within 6 months post-intervention. | Pediatricians | Help-seeking behavior was measured by the use of mental health care services, defined as making at least one appointment (versus no appointment) by the 6-month follow-up. Patients on waiting lists were counted as having a positive outcome. | No significant difference between arms |
| Sekhar et al., 2022 | USA | High school students (adolescents and young adults) | *n* = 12,909 Females: 46.1% | Median age = 16 years Range: 13–21 years | Universal screening | School | Initiation of recommended mental health services/ treatment (≥1 SAP – recommended service: school/ community/SAP liaison-based; after SAP assessment) assessed post-identification during school year | Automated | Help-seeking behavior was assessed by determining if participants initiated at least one recommended Student Assistance Program (SAP) treatment or service, including school-based, community-based or SAP liaison services. | Intervention significantly increased treatment seeking behaviors (OR = 4.0-fold higher, 95% CI 2.0–7.9). |
| **Primary Outcome: Help-seeking Intention** | | | | | | | | | | |
| Amsalem, Jankowsk et al., 2023 | USA | Study 1: Young adults Study 2: adolescents | Study 1, *n* = 895 Females: 428 (48%) Males: 467 (52%) Study 2, *n* = 637 Females: 309 (49%) Males: 328 (51%) | Mean [SD] Study 1 = 23.9 ± 3.7 years (range 18–30) Study 2 = 17.1 ± 1.1 years (range 14–18) | Social contact-based video interventions. | Community | Treatment-seeking intention (likelihood to seek help from 10 sources, e.g., mental health professional/ parent/helpline) assessed during baseline/post-intervention | Online | Help-seeking intention was measured using the General Help-Seeking Questionnaire (GHSQ) | Intervention significantly increased treatment-seeking intentions. Cohen's *d* effect sizes ranged from 0.14 to 0.33. Composite score not reported |
| Wiljer et al., 2016 | Canada | Students | *n* = 481 Females: 378 (78.4%) Males: 91 (18.9%) Nonbinary: 12 (2.3%) | Mean [SD] = 23.05 ± 3.25 years Range: 17–29 years | Promotion of help-seeking resources | University | Formal MH services (e.g., mental health professionals/ helplines/doctors); informal (e.g., friends/family/religious leaders); intention/behavior both via GHSQ/AHSQ assessed baseline/3 months/6 months self-report. | Online | Help-seeking intention was measured using the General Help-Seeking Questionnaire (GHSQ) | No significant difference between the arms |

**Table 4.** Mapping of intervention components with outcomes and reported effectiveness

| Author, Year | Mental health literacy/psychoeducation (building awareness to recognize symptoms and risk factors to reduce stigma related to the condition. | Providing information about support services and treatment options (practical guidance on how to access care including self-help resources/helplines/ directories of local MH services/ professionals. | Training individuals to identify and refer those at risk for mental health conditions | Training individuals to raise awareness about mental health and help-seeking resources | Training delivery agents in MI-based strategies | MI-based case management | Motivational enhancement | Cognitive Behavioral Therapy for treatment-seeking | Group counseling | Peer-led Support (e.g., online support groups facilitated by peers with lived experience). | Using a clinician-supported mHealth app |
|---|---|---|---|---|---|---|---|---|---|---|---|
| | Mental health Literacy | | Training and Skills Development | | | MI Strategies | | Counseling-based strategies and support | | | |
| Achterbergh et al., 2021 | | | | | • | | | | | | |
| Aisenberg-Shafran and Shturm, 2022 | | | | | | | | | | | |
| Amsalem et al., 2021 | | | | | | | | | | | |
| Amsalem, Lazarov et al., 2022 | | | | | | | | | | | |
| Amsalem, Jankowsk et al., 2023 | | | | | | | | | | | |
| Amsalem, Wall et al., 2022 | | | | | | | | | | | |
| Amsalem, Wall et al., 2023 | | | | | | | | | | | |
| Calear et al., 2022 | | | | • | | | | | | | |
| Casañas et al., 2022 | • | | | | | | | | | | |
| Coleman et al., 2019 | | | • | | | | | | | | |
| Conceição et al., 2022 | • | | | | | | | | | | |
| Fernandez et al., 2022 | | | | | | | | | | | |
| Finitsis et al., 2022 | • | | | | • | | | | | | |
| Gilgoff et al., 2023 | • | | | | | | | | | | |

*(Continued)*

**Table 4.** (*Continued*)

| Author, Year | Mental health literacy/ psychoeducation (building awareness to recognize symptoms and risk factors to reduce stigma related to the condition. | Providing information about support services and treatment options (practical guidance on how to access care including self-help resources/helplines/ directories of local MH services/ professionals. | Training individuals to identify and refer those at risk for mental health conditions | Training individuals to raise awareness about mental health and help-seeking resources | Training delivery agents in MI-based strategies | MI-based case management | Motivational enhancement | Cognitive Behavioral Therapy for treatment-seeking | Group counseling | Peer-led Support (e.g., online support groups facilitated by peers with lived experience). | Using a clinician-supported mHealth app |
|---|---|---|---|---|---|---|---|---|---|---|---|
| | Mental health Literacy | | Training and Skills Development | | | MI Strategies | | Counseling-based strategies and support | | | |
| Grupp Phelan et al., 2019 | | | | | • | | | | | | |
| Han et al., 2023 | • | • | | | | | | | | | |
| Hollar and Siegel, 2020 | | | | | | | | | | | |
| Hollar and Siegel, 2023 | | | | | | | | | | | |
| Holt et al., 2017 | | | | | • | | | | | | |
| Hussain et al., 2022 | | | | | | | | | | | |
| Hussain and Alhabash, 2020 | | | | | | | | | | | |
| Ilgen et al., 2022 | • | | | | | | | | | | |
| Jordans et al., 2020 | | | | | | | | | | | |
| Kerman et al., 2023 | | | | | | | | | | | |
| King et al., 2018 | | | | | | | | | | | |
| King et al., 2022 | | | | | | | | | | | |
| King et al., 2023 | • | • | | | | | | | | | |
| Kruger et al., 2022 | | | | | | | | | | | |
| Lee et al., 2023 | | | | | | | | | | | |
| Link et al., 2020 | • | | | | | | | | | | |
| Lubman et al., 2020 | • | | • | | | | | | | | |
| Lustig et al., 2023 | | | | | | | | | | | |

**Table 4.** (*Continued*)

| Author, Year | Mental health literacy/psychoeducation (building awareness to recognize symptoms and risk factors to reduce stigma related to the condition.) | Providing information about support services and treatment options (practical guidance on how to access care including self-help resources/helplines/ directories of local MH services/ professionals.) | Training individuals to identify and refer those at risk for mental health conditions | Training individuals to raise awareness about mental health and help-seeking resources | Training delivery agents in MI-based strategies | MI-based case management | Motivational enhancement | Cognitive Behavioral Therapy for treatment-seeking | Group counseling | Peer-led Support (e.g., online support groups facilitated by peers with lived experience). | Using a clinician-supported mHealth app |
|---|---|---|---|---|---|---|---|---|---|---|---|
| | Mental health Literacy | | Training and Skills Development | | | MI Strategies | | Counseling-based strategies and support | | | |
| Martin et al., 2020 | | | | | | | | | | | |
| Mason et al., 2022 | | • | | | | | | | | | |
| Milner et al., 2018 | • | • | | | | | | | | | |
| Mori et al., 2022 | • | | | | | | | | | | |
| Nickerson et al., 2020 | • | • | | | | | | | | | |
| O'Dea et al., 2021 | • | • | | | | | | | | | |
| Parikh et al., 2021 | • | | | | | | | | | | |
| Possemato et al., 2023 | | | | | | | | | | | • |
| Reinauer et al., 2021 | | | | | • | | | | | | |
| Ries et al., 2022 | • | | | | | | | | | | |
| Scott et al., 2023 | | | | | | • | | | | | |
| Sekhar et al., 2022 | | | | | | | | | | | |
| Seidman et al., 2022 | | | | | | | | | • | | |
| Shafran et al., 2019 | • | | | | | | | | | | |
| Smith et al., 2023 | | | | | | | | | | | |
| Stecker et al., 2023 | | | | | | | | • | | | |
| Straszewski and Siegel, 2018 | | | | | | | | | | | |

(*Continued*)

| Author, Year | Mental health literacy/psychoeducation (building awareness to recognize symptoms and risk factors to reduce stigma related to the condition.) | Providing information about support services and treatment options (practical guidance on how to access care including self-help resources/helplines/ directories of local MH services/ professionals.) | Training individuals to identify and refer those at risk for mental health conditions | Training individuals to raise awareness about mental health and help-seeking resources | Training delivery agents in MI-based strategies | MI-based case management | Motivational enhancement | Cognitive Behavioral Therapy for treatment-seeking | Group counseling | Peer-led Support (e.g., online support groups facilitated by peers with lived experience). | Using a clinician-supported mHealth app |
|---|---|---|---|---|---|---|---|---|---|---|---|
| | Mental health Literacy | | Training and Skills Development | | | MI Strategies | | Counseling-based strategies and support | | | |
| Tay, 2022 | • | • | | | | | | | | | |
| Till et al., 2024 | | | | | | | | | | | |
| Tobias et al., 2022 | • | | | | | | • | | | | |
| Ugarte et al., 2023 | • | | | | | | | | | • | |
| Wiljer et al., 2016 | | • | | | | | | | | | |

| Author, Year | Writing tasks to induce specific mental or emotional states | Facilitated group discussions on mental health. | Interviews & news items | Public service announcement videos | Magazine articles | Documentaries | Brief videos | Social contact/ interactions | Screening at schools/ university | Screening at facility level | Automated/ algorithm screening |
|---|---|---|---|---|---|---|---|---|---|---|---|
| | Writing and Discussion | | Media | | | | | Social Contact | Screening | | |
| Achterbergh et al., 2021 | | | | | | | | | | | |
| Aisenberg-Shafran and Shturm, 2022 | | | | | | | | | | | |
| Amsalem et al., 2021 | | | | | | | • | • | | | |
| Amsalem, Lazarov et al., 2022 | | | | | | | • | • | | | |
| Amsalem, Jankowsk et al., 2023 | | | | | | | • | • | | | |
| Amsalem, Wall et al., 2022 | | | | | | | • | • | | | |
| Amsalem, Wall et al., 2023 | | | | | | | • | • | | | |
| Calear et al., 2022 | | | | | | | | | | | |
| Casañas et al., 2022 | | | | | | | | • | | | |
| Coleman et al., 2019 | | | | | | | | | | | |
| Conceição et al., 2022 | | | | | | | | • | | | |
| Fernandez et al., 2022 | | | | | • | | • | | | | |
| Finitsis et al., 2022 | | | | | | | | | | | |

(Continued)

**Table 4.** (*Continued*)

| Author, Year | Writing tasks to induce specific mental or emotional states | Facilitated group discussions on mental health. | Interviews & news items | Public service announcement videos | Magazine articles | Documentaries | Brief videos | Social contact/ interactions | Screening at schools/ university | Screening at facility level | Automated/ algorithm screening |
|---|---|---|---|---|---|---|---|---|---|---|---|
| | Writing and Discussion | | Media | | | | | Social Contact | Screening | | |
| Gilgoff et al., 2023 | | | | | | | | | | | |
| Grupp Phelan et al., 2019 | | | | | | | | | | | |
| Han et al., 2023 | | • | | | | | | | | | |
| Hollar and Siegel, 2020 | • | | | | | | | | | | |
| Hollar and Siegel, 2023 | • | | | | | | | | | | |
| Holt et al., 2017 | | | | | | | | | | | |
| Hussain et al., 2022 | | | | • | | | | | | | |
| Hussain and Alhabash, 2020 | | | | • | | | | | | | |
| Ilgen et al., 2022 | | | | | | | | | | | |
| Jordans et al., 2020 | | | | | | | | | | • | |
| Kerman et al., 2023 | | | | | | | | | | | |
| King et al., 2018 | | | | | | • | | • | | | |
| King et al., 2022 | | | | | | | | | • | | |
| King et al., 2023 | | | | | | | • | • | | | |
| Kruger et al., 2022 | | | | | | | • | • | | | |
| Lee et al., 2023 | | | | | | | | | | | |
| Link et al., 2020 | | | | | | | | • | | | |
| Lubman et al., 2020 | | | | | | | | | | | |
| Lustig et al., 2023 | | | | | | | | | • | | |
| Martin et al., 2020 | | | | | | | | | | | |
| Mason et al., 2022 | | | | | | | | | | | |
| Milner et al., 2018 | | | | | | | | | | | |
| Mori et al., 2022 | | | | | | | | • | | | |
| Nickerson et al., 2020 | | | | | | | | • | | | |
| O'Dea et al., 2021 | | | | | | | | | | | |
| Parikh et al., 2021 | | | | | | | | | | | |
| Possemato et al., 2023 | | | | | | | | | | | |
| Reinauer et al., 2021 | | | | | | | | | | | |

**Table 4.** (*Continued*)

| Author, Year | Writing tasks to induce specific mental or emotional states | Facilitated group discussions on mental health. | Interviews & news items | Public service announcement videos | Magazine articles | Documentaries | Brief videos | Social contact/ interactions | Screening at schools/ university | Screening at facility level | Automated/ algorithm screening |
|---|---|---|---|---|---|---|---|---|---|---|---|
| | Writing and Discussion | | Media | | | | | Social Contact | Screening | | |
| Ries et al., 2022 | | | | | | | | | | | |
| Scott et al., 2023 | | | | | | | | | | • | |
| Sekhar et al., 2022 | | | | | | | | | • | | • |
| Seidman et al., 2022 | • | | | | | | | | | | |
| Shafran et al., 2019 | | | | | | | | | | | |
| Smith et al., 2023 | | | | | | | | | | | |
| Stecker et al., 2023 | | | | | | | | | | | |
| Straszewski and Siegel, 2018 | • | | | | | | | | | | |
| Tay, 2022 | | | | | | | | | | | |
| Till et al., 2024 | | | • | | | | | | | | |
| Tobias et al., 2022 | | | | | | | | | | | |
| Ugarte et al., 2023 | | | | | | | | | | | |
| Wiljer et al., 2016 | | | | | | | | | | | |

| Author, Year | Health system level support | Referral to treatment | Providing self-help resources | Mindfulness-Based Intervention for Seniors (MBIS) | Alternative labels used for mental health providers (e.g., clinical psychologist vs. mind coach) | Symptom monitoring | Personalized/ tailored feedback | Financial incentives | Encouraging social connection with others | Behavior | Intention | Attitude |
|---|---|---|---|---|---|---|---|---|---|---|---|---|
| | Other strategies | | | | | | | | | Help-seeking Outcomes | | |
| Achterbergh et al., 2021 | | | | | • | | | | | ▼ | | |
| Aisenberg-Shafran and Shturm, 2022 | | | | • | | | | | | | | ▲ |
| Amsalem et al., 2021 | | | | | | | | | | | ▲ | |
| Amsalem, Lazarov et al., 2022 | | | | | | | | | | | ▲ | |
| Amsalem, Jankowsk et al., 2023 | | | | | | | | | | | ▲ | |
| Amsalem, Wall et al., 2022 | | | | | | | | | | | | ▲ |
| Amsalem, Wall et al., 2023 | | | | | | | | | | | ▲ | |
| Calear et al., 2022 | | | | | | | | | • | ▼ | ▼ | |

(*Continued*)

**Table 4.** (*Continued*)

| Author, Year | Health system level support | Referral to treatment | Providing self-help resources | Mindfulness-Based Intervention for Seniors (MBIS) | Alternative labels used for mental health providers (e.g., clinical psychologist vs. mind coach) | Symptom monitoring | Personalized/ tailored feedback | Financial incentives | Encouraging social connection with others | Help-seeking Outcomes Behavior | Intention | Attitude |
|---|---|---|---|---|---|---|---|---|---|---|---|---|
| | | | | | Other strategies | | | | | Behavior | Intention | Attitude |
| Casañas et al., 2022 | | | | | | | | | | ▼ | | |
| Coleman et al., 2019 | | | | | | | | | | ▲ | ▲ | |
| Conceição et al., 2022 | | | | | | | | | | | | ▲ |
| Fernandez et al., 2022 | | | | | | | | | | | ▼ | |
| Finitsis et al., 2022 | | | | | | | | | | ▲ | | |
| Gilgoff et al., 2023 | | | • | | | | | | | ▲ | | |
| Grupp Phelan et al., 2019 | | | | | | | | | | ▼ | | |
| Han et al., 2023 | | | | | | | | | | | ▲ | ▼ |
| Hollar and Siegel, 2020 | | | | | | | | | | | ▲ | |
| Hollar and Siegel, 2023 | | | | | | | | | | | ▲ | ▲ |
| Holt et al., 2017 | | | | | | | | | | ▲ | | |
| Hussain et al., 2022 | | | | | | | | | | | ▼ | ▲ |
| Hussain and Alhabash, 2020 | | | | | | | | | | | ▼ | ▼ |
| Ilgen et al., 2022 | | | | | | | | | | ▼ | | |
| Jordans et al., 2020 | | | | | | | | | | ▲ | | |
| Kerman et al., 2023 | | | | | | | • | | | ▼ | | |
| King et al., 2018 | | | | | | | | | | | ▲ | |
| King et al., 2022 | | • | | | | | • | | | ▼ | | |
| King et al., 2023 | | | | | | | | | | | ▼ | |
| Kruger et al., 2022 | | | | | | | | | | | ▲ | |
| Lee et al., 2023 | • | | | | | | | | | ▼ | | |
| Link et al., 2020 | | | | | | | | | | ▲ | | |
| Lubman et al., 2020 | | | | | | | | | | ▼ | | |
| Lustig et al., 2023 | | • | | | | | | | | ▼ | | |
| Martin et al., 2020 | | | | | | | | | | ▲ | ▲ | ▲ |
| Mason et al., 2022 | | | | | | | | | | | ▲ | |
| Milner et al., 2018 | | | | | | | • | | | | | ▼ |
| Mori et al., 2022 | | | | | | | | | | ▼ | | |

**Table 4.** (*Continued*)

| Author, Year | Health system level support | Referral to treatment | Providing self-help resources | Mindfulness-Based Intervention for Seniors (MBIS) | Alternative labels used for mental health providers (e.g., clinical psychologist vs. mind coach) | Symptom monitoring | Personalized/ tailored feedback | Financial incentives | Encouraging social connection with others | Help-seeking Outcomes Behavior | Intention | Attitude |
|---|---|---|---|---|---|---|---|---|---|---|---|---|
| | | | | | Other strategies | | | | | | | |
| Nickerson et al., 2020 | | | | | | | | | | ▲ | ▲ | |
| O'Dea et al., 2021 | | | | | | | | | | | ▲ | |
| Parikh et al., 2021 | | | | | | | | | | ▲ | | |
| Possemato et al., 2023 | | | | | | | | | | ▲ | | |
| Reinauer et al., 2021 | | | | | | | | | | ▼ | | |
| Ries et al., 2022 | | | | | | | | | | ▲ | | |
| Scott et al., 2023 | | • | | | | | | | | ▲ | | |
| Sekhar et al., 2022 | | | | | | | | | | ▲ | | |
| Seidman et al., 2022 | | | | | | | | | | | ▼ | ▲ |
| Shafran et al., 2019 | | | • | | | • | | | | ▼ | | |
| Smith et al., 2023 | | | | | • | | | | | | ▲ | |
| Stecker et al., 2023 | | | | | | | | | | ▲ | | |
| Straszewski and Siegel, 2018 | | | | | | | | | | | ▲ | |
| Tay, 2022 | | | | | | | | | | | ▼ | ▲ |
| Till et al., 2024 | | | | | | | | | | | ▲ | |
| Tobias et al., 2022 | | | | | | | | | | ▲ | ▲ | ▲ |
| Ugarte et al., 2023 | | | | | | | | | | ▲ | | |
| Wiljer et al., 2016 | | | | | | | | | | | ▼ | |

*Note:* The different colors indicate the level of risk of bias: green indicates a low risk of bias, yellow indicates a moderate risk of bias and red indicates a high risk of bias.
▲ = statistically significant effect in the intervention group; ▼ = statistically non-significant effect in the intervention group.

1 month and persisting (OR = 1.37) at 12 months ($p$ = 0.05) (Stecker et al., 2023). "Man Therapy" ($n$ = 354 men at suicide risk) used masculine humor and language in their gender-responsive psychoeducation and self-help resources to align help-seeking with masculine autonomy, increasing professional help-seeking (OR = 1.55, $p$ = 0.049) (Gilgoff et al., 2023). "Tell Your Story" ($n$ = 103 refugee men, Arabic/Farsi/Tamil-speaking backgrounds) combined psychoeducation, video-based social contact (refugee men describing symptom experiences and help-seeking journeys) and cognitive reappraisal, which resulted in a specific reduction in self-stigma related to help-seeking itself ($d$ = 0.42). Participants accessed approximately 0.69 additional help-seeking sources at follow-up ($B$ = 0.69, 95% CI 0.19–1.18, $p$ = 0.007), including informal (friends, family and community leaders) and formal sources (GPs, mental health professionals) (Nickerson et al., 2020).

Six studies also reported secondary outcomes; four of which were effective in decreasing help-seeking stigma (Martin et al., 2020; Nickerson et al., 2020; Ries et al., 2022; Gilgoff et al., 2023) and five were effective in improving knowledge about mental health conditions (Coleman et al., 2019; Link et al., 2020; Parikh et al., 2021; Ries et al., 2022; Gilgoff et al., 2023).

### Interventions targeting help-seeking intentions

Of 26 studies targeting help-seeking intentions across settings, interventions in 19 (73.1%) proved effective.

#### *University/school settings ($n$ = 4/9 effective studies)*

Resource promotion with MHL: Across educational settings, intention change was achieved through resource promotion (providing information about support services and treatment options, such as practical guidance on how to access care, including self-help resources/helplines/directories of local MH services/ professionals), often paired with mental health literacy (MHL) delivered in different formats. Mason et al. (2022, $n$ = 133 undergraduates) tested infographics vs. videos promoting campus services and found the infographic intervention significantly increased help-seeking intentions (M = 3.52 vs. M = 3.05, $p$ = 0.035). Multi-component customized MHL + social contact + resource promotion for college athletes (Martin et al., 2020, $n$ = 107) significantly increased intentions for both eating pathology and general mental health help-seeking ($\eta^2$ = 0.09, $p$ = 0.003).

#### *Community-based settings ($n$ = 14/16 effective studies)*

Social contact: The most prominent strategy involved video-based social contact interventions, particularly tailored to viewer characteristics (e.g., profession, gender and cultural congruence). Seven studies were effective in this cluster: healthcare workers (Amsalem, Lazarov, et al., 2022; Amsalem, Wall, et al., 2023), essential workers (Amsalem and Martin, 2022), male construction workers (King et al., 2018), young adults and adolescents (Kruger et al., 2022; Amsalem, Jankowski, et al., 2023) and refugee men (Nickerson et al., 2020).

Integrated psychoeducation + behavioral engagement: Adult and youth interventions combining mental health education with active skill-building or discussion/interactive activities showed significant intention gains. Brief single-session psychoeducation + CBT + MI-based strategies for adults with social anxiety (Tobias et al., 2022, $n$ = 241, $d$ = 0.18, $p$ = 0.06) significantly increased treatment-seeking intentions. Psychoeducation

+ facilitated group discussion + social contact (tailored to match audience characteristics) + resource promotion for refugee men (Nickerson et al., 2020, $d$ = 0.27, $p$ = 0.027) and SGM youth (Han et al., 2023, $\Delta$ = 0.17–0.19, $p$ = 0.013–0.018) similarly increased help-seeking intentions.

Self-distancing writing tasks: Two studies testing emotion-regulation writing interventions targeting adults with depressive symptoms demonstrated significant improvements in help-seeking intentions. Hollar and Siegel (2020) conducted a series of three online studies (combined $n$ = 632) in which participants completed writing tasks designed to induce self-distancing while imagining seeking help from the perspective of their ideal future self, significantly increasing help-seeking intentions, with the largest effect observed in Study 3 ($n$ = 574, $\eta p^2$ = 0.11, $p$ = 0.011). Similarly, writing prompts designed to induce positive emotional states ($n$ = 136) showed significant intention gains ($\eta p^2$ = 0.04, $p$ = 0.032, M = 4.09) (Straszewski and Siegel, 2018).

Among the studies ($n$ = 7) that reported secondary outcomes, five interventions were effective in decreasing stigma (Hollar and Siegel, 2020; Nickerson et al., 2020; Mason et al., 2022; Seidman et al., 2022; Amsalem, Wall, et al., 2023), one was effective in increasing MHL (Han et al., 2023) and one was effective in increasing gatekeeper efficacy (Coleman et al., 2019).

### Interventions targeting help-seeking attitudes

Of the 12 studies targeting attitudes, 10 (83.3%) were effective. Most effective strategies moved beyond providing "facts" to using social contact either via video or peer narratives (Martin et al., 2020; Amsalem, Wall, et al., 2022; Han et al., 2023).

#### *Interventions tested in a school/university setting*

MHL with interactive/social-contact elements: Universal MHL interventions that mapped symptom-treatment literacy to make help-seeking predictable and evidence-based (Martin et al., 2020; Conceição et al., 2022; Tay, 2022) succeeded in attitudinal change. Conceição et al. (2022, $n$ = 626 first-year university students in Portugal) delivered online psychoeducation combined with social contact and interactive elements (like quizzes), achieving significant attitude improvement ($p$ < .001, Cohen's f = 0.04). MHL sessions addressing barriers to treatment, such as cost, side effects, intervention modalities and medication myths about antidepressants and antipsychotics demonstrated significantly greater recognition of these interventions as viable treatment options ($n$ = 174; U = 2,774.50–2,799, $p$ < .001) (Tay, 2022).

Among the studies ($n$ = 5) that reported secondary outcomes related to stigma in help-seeking and MHL, two interventions were effective in decreasing self-stigma related to help-seeking (Martin et al., 2020; Seidman et al., 2022) and two were effective in increasing MHL (Tay, 2022; Han et al., 2023).

### Risk of bias

Most RCTs assessed with the RoB 2 tool were of moderate quality ($n$ = 23), with common concerns in result selection, outcome measurement and missing data. Bias levels varied due to insufficient data, often resulting in the "no information" option. Detailed assessments are in Supplementary Appendix B.

## Discussion

Despite our searches being conducted only 7 years after the Xu et al. (2018) review, we identified 54 new studies published in this period that tested several help-seeking interventions, highlighting a growing interest in addressing demand-side barriers. Of the included studies, 36 reported interventions were effective in improving one or more outcome measures related to help-seeking.

Help-seeking interventions did not work through a single universal mechanism. Instead, their effectiveness depended on matching the intervention strategies to the main barrier faced by a given population (e.g., lack of awareness, identity-related stigma or practical access problems) for the outcomes targeted (behaviors, intentions and attitudes). For instance, brief digital social contact tended to work best to increase help-seeking intentions in general-population samples with low perceived need; more intensive, motivationally focused work tended to trigger behavior change in clinical populations already in contact with care; and school-based referral models were most effective to trigger help-seeking behavior when adolescents could recognize a problem but could not independently navigate access to services.

Clinical or at-risk populations required higher-intensity engagement addressing internal barriers (ambivalence, competing priorities) rather than external barriers (stigma, awareness) to elicit behavioral help-seeking. Motivational Interviewing (MI) and dedicated linkage models succeed by resolving motivational and logistical barriers simultaneously within the healthcare settings in increasing help-seeking behaviors (Holt et al., 2017; Scott et al., 2023). Health-system integrated dedicated staff (e.g., primary care linkage workers or nurses) using MI-based strategies to resolve patient-level barriers (ambivalence about efficacy, time/cost constraints, stigma) during extended engagement eliminated post-intention logistical friction, translating into behavioral change. This works for clinical populations who have overcome awareness barriers but face internal motivation barriers, such as "Will treatment work?" or "Can I manage alongside responsibilities?" (Bischof et al., 2021). MI addresses these through collaborative exploration of ambivalence. Dedicated linkage staff eliminates appointment scheduling, transportation and treatment selection friction that high-risk populations often struggle to navigate independently. However, these seem to be effective with adult clinical populations with acknowledged symptoms (SUD, postpartum distress, chronic mental illness) and may not be useful for those in crisis or denial of symptoms.

Despite the diversity of intervention approaches across educational, healthcare and community settings, several overarching mechanisms emerged associated with intervention success (McLaren et al., 2023). Interventions targeting behavior change, effectiveness emerged through a consistent pattern. Whether via technology-enabled screening, peer training or in-person sensitization, the most effective interventions shared a critical outcome of removing friction between awareness, access and action, though they achieved this through mechanistically distinct approaches. Some interventions addressed the barrier of problem recognition (screening, gatekeeper training), while others addressed barriers of access to knowledge (sensitization with embedded pathways) or psychological ambivalence (peer-delivered MI). Across all settings, effective interventions matched their mechanisms to specific barriers to initiate behavior change, that is, once young people or patients recognized a need, the path to professional support was immediate, accessible and socially legitimized.

Adolescents with emerging symptoms require immediate, frictionless translation from symptom recognition to professional contact. School-based settings that enable direct or self-identification and immediate referral tend to eliminate this decision-making friction (Link et al., 2020; Parikh et al., 2021; Sekhar et al., 2022). This creates a "recognition-to-action pathway" without a choice burden, where students learned they have a problem and know exactly how and where to access help. A notable mention of intervention is where students received mental health knowledge via video and discussion, then were given clear referral instructions (meet a counselor directly, drop a box outside their room or teacher referral), all on school premises, eliminating search and transportation barriers (Parikh et al., 2021). Adolescents may have low symptom awareness combined with high social barriers to help-seeking, thus embedding identification and referral within the school day eliminates transportation, scheduling and parental navigation requirements. This seems effective for school-enrolled adolescents with emerging (not chronic) symptoms and accessible mental health services.

This review identifies a pronounced intention-behavior gap where, while intentions were effectively changed in 19 of 26 studies (73.1%) and actual behaviors were changed in only 16 of 29 studies (55.2%). This aligns with established literature (Rickwood and Thomas, 2012) suggesting that while social contact and MHL can successfully foster a "willingness" to seek help, the transition to "action" is frequently hampered by structural and psychological "friction." Interventions that successfully bridged this gap such as the RMC-PC model (Scott et al., 2023) in the United States or lay-counselor sensitization (Parikh et al., 2021) in India did so by proactively removing the burden of navigation from the individual to institutional systems, and addressing psychological (stigma, ambivalence), structural (navigation, appointment-seeking, treatment selection) and capacity barriers (time, transportation, competing priorities).

Intervention categories that were most commonly found to be effective were psychoeducation/MHL ($n = 13$), promotion of help-seeking resources ($n = 11$), social contact ($n = 7$), MI ($n = 4$) and screening ($n = 3$). Psychoeducation/MHL, social contact and help-seeking resource promotions were most effective when combined, improving all three outcomes in both the general population and those with mental health conditions. This aligns with earlier research showing that insufficient knowledge and stigma hinder individuals from seeking help (Mojtabai et al., 2011; Schnyder et al., 2017; Xu et al., 2018). Social contact interventions especially worked for help-seeking intentions in the general population (Thornicroft et al., 2016; Xu et al., 2018) and MI-based strategies combined with psychoeducation reported effectiveness for improving help-seeking behaviors in at-risk populations and people with mental health conditions (Xu et al., 2018). MHL/psychoeducation was also found to be effective in vulnerable populations like refugees and sexual/gender minorities (Nickerson et al., 2020; Han et al., 2023).

Previous reviews show that help-seeking interventions improve formal help-seeking behaviors in individuals with, or at risk of, mental health conditions (Xu et al., 2018), but are less effective for the general population (Gulliver et al., 2012; Xu et al., 2018). Mental health literacy improved help-seeking attitudes ($d = .12$ to $.53$) but not behaviors ($d = -.01, .02$); evidence for other interventions was limited (Gulliver et al., 2012). Digital interventions with active personal involvement and social contact have been shown to effectively boost help-seeking intentions and may be more cost-effective than face-to-face methods (Evans-Lacko et al., 2022). Studies also indicate that multi-component interventions are generally more effective for improving help-seeking outcomes than single-component interventions (Xu et al., 2018; Evans-Lacko et al., 2022; van den Broek et al., 2023).

Low-intensity digital interventions, particularly social contact videos, could achieve high population-level impact on help-seeking intentions by making help-seeking appear less stigmatized and acceptable, without requiring additional clinical infrastructure. Most community-based studies targeting intentions that were social-contact-based and delivered online were effective (14 of 16; 87.5%). These effects were often stronger when the people shown in the videos resembled the target audience in age, gender, occupation or cultural background (i.e., demographic tailoring), because this reduces the sense that "people like me do not seek help" (King et al., 2018; Nickerson et al., 2020; Amsalem et al., 2021; Amsalem et al., 2022; Amsalem, Jankowski, et al., 2023). This aligns with social contact theory, which posits that vicarious contact (e.g., via video testimonials) reduces stigma by disconfirming stereotypes and fostering ingroup identification with help-seekers, thereby increasing perceived normativity of help-seeking behavior (Allport, 1954; Corrigan et al., 2013; Clement et al., 2012). Delivering these materials online also removes practical steps such as signing up for a session or attending a group, which is particularly important for people who do not yet feel a strong personal need for care (yet).

Studies indicate that mechanisms that act on intentions effectively promote behavior change (Xiao et al., 2019; Amsalem, Wall, et al., 2022). These interventions can facilitate immediate changes in treatment-seeking behaviors by providing online counselor referrals. This underscores the potential of social contact-based video interventions as cost-effective, scalable strategies for promoting help-seeking intentions, attitudes and reducing stigma, particularly in resource-constrained settings (Evans-Lacko et al., 2012, 2022). Once produced, brief social contact videos can be disseminated to thousands while maintaining fidelity to the "active ingredient" (e.g., the peer narrative) via social media, clinical portals or websites at near-zero marginal cost per user (Clement et al., 2012; Lehtimaki et al., 2021; Buntrock, 2024; Amsalem et al., 2025). It also does not require highly trained professionals for delivery, which is a significant availability and economic bottleneck in traditional mental health service expansion. Future research should explore whether incorporating behavior change elements into social-contact video interventions can effectively translate improved intentions and attitudes into actual behavior change.

While most of the evidence originates from high-income countries (HICs), the two LMIC studies provide critical signals for low-resource implementation. Jordans et al. (2020, Nepal, *n* = 286,838) demonstrated that community health volunteers conducting proactive screening achieved significant treatment registration by externalizing symptom recognition, essential in contexts with high social barriers to self-disclosure. Parikh et al. (2021, India, *n* = 3,587) achieved significantly better outcomes through lay counselor-delivered classroom sensitization with immediate on-site referral options, highlighting that MHL alone fails without embedded physical access in resource-constrained settings. However, this limited evidence (two studies from South Asia) precludes definitive LMIC conclusions. Both succeeded through friction removal (externalized recognition, proximal access), suggesting context-matched structural solutions may generalize, but a lack of geographic diversity (no sub-Saharan Africa, Latin America) and unmeasured cost-effectiveness evidence prevent broad recommendations. For LMIC contexts, digital social contact appears promising as a scalable "gateway" (proven in HICs), while task-shared linkage models (volunteers and lay counselors) could function as essential "bridges" where specialist supply is limited.

One study that reported on mediation analysis found that positive emotions and attitudes together mediated the impact of message appeal on help-seeking intentions, suggesting a serial mediation effect (Hussain and Alhabash, 2022). However, most studies did not examine mediation analyses, limiting insights into optimizing these interventions.

Multi-component interventions were more effective for multiple help-seeking outcomes. Two studies (Martin et al., 2020; Tobias et al., 2022) that effectively improved all three help-seeking outcomes combined psychoeducation, stigma reduction, resource promotion, single-session CBT and motivational strategies, and focused on specific clinical conditions (Xu et al., 2018; van den Broek et al., 2023).

## Limitations

The inherent complexity of help-seeking as a construct, characterized by the lack of consensus on definitions and measurements as highlighted by Rickwood and Thomas (2012), contributed significantly to the heterogeneity observed in our results. This diversity in how studies operationalized "help," ranging from clinical appointments to online peer engagement, precluded a meta-analysis and underscores the need for the use of more standardized research frameworks in the future. Limited data and outcome variability reported by the included studies prevented meaningful conclusions on implementation outcomes. In an attempt to thoroughly examine the research landscape and limited studies in the area, the quality assessment of studies did not impact the weight given to them in the narrative synthesis. We did not search gray literature or publications in languages other than English, potentially biasing our results. The wide variation in sample sizes, with some as small as 24 participants, limits the precision and generalizability of our findings. Most studies did not explore the mechanisms of change, making it hard to identify which particular components affect help-seeking outcomes. Additionally, limited evidence from LMICs may hinder the applicability of findings to low-resource settings. LMIC-based evidence is restricted to just two countries from South Asia (Jordans et al., 2020; Parikh et al., 2021). Patterns suggest context-matched friction removal is effective, but geographic and systemic diversity across LMICs prevents definitive conclusions. Those implementing in LMIC contexts should conduct local formative research and cost-effectiveness analysis before scale-up.

## Conclusion

Our findings align with previous research, confirming the effectiveness of MHL or psychoeducation, MI-based strategies and social contact interventions across multiple help-seeking outcomes (Gulliver et al., 2012; Xu et al., 2018; Velasco et al., 2020; Evans-Lacko et al., 2022). Social contact videos emerged to be consistently effective at increasing help-seeking intentions and attitudes.

Future research should focus on examining help-seeking outcomes in alignment with specific theories like TPB, exploring mechanisms of change and using standardized measures (Rickwood and Thomas, 2012). This approach will improve our understanding of how help-seeking can be enhanced and what mediates the relationship between intentions, attitudes and behaviors.

**Open peer review.** To view the open peer review materials for this article, please visit http://doi.org/10.1017/gmh.2026.10183.

**Supplementary material.** The supplementary material for this article can be found at http://doi.org/10.1017/gmh.2026.10183.

**Data availability statement.** All data analyzed in this study are derived from publicly available sources and are fully reported within the article and its Supplementary Materials. Extraction sheets used in this review are available from the corresponding author upon reasonable request.

**Acknowledgments.** We thank Ankur Garg (AG) for their help in reviewing and resolving conflicts throughout the abstract and full-text screening process. We also extend our thanks to Neo Hou (NH) for assisting with data extraction and cleaning the data in the extraction sheet.

**Author contribution.** Conceptualization: A.N.; Data curation: S.B., S.S.I.; Formal analysis: S.B., B.B.; Investigation: S.B., S.S.I., M.P., N.N.; Methodology: A.N.; Supervision: Y.G., R.V., D.R.S., V.P.; Validation: S.B., Y.G., R.V., D.R.S., V.P., A.N.; Visualization: S.B.; Writing – original draft: S.B., B.B., S.S.I.; Writing – review and editing: S.B., B.B., Y.G., R.V., D.R.S., V.P., S.S.I., M.P., N.N., A.N. S.I. drafted an initial version of the manuscript, which was substantially reworked by S.B. to incorporate new data and refine the analysis. S.B. then continued developing revised drafts in collaboration with B.B. All authors contributed to reviewing the revised drafts before finalizing the submitted version of the study.

**Financial support.** This study was conducted as a part of the formative research for the IMPRESS trial, which aims to assess the effectiveness and cost-effectiveness of a community intervention in enhancing access to care and improving clinical outcomes for depression. The trial has been funded through a grant from the National Institute of Mental Health (NIMH), USA (Grant number R01MH115504). The funder had no role in the design, conduct, analysis or reporting of this systematic review and did not influence its findings or conclusions in any way.

**Competing interests.** The authors declare that there are no competing interests.

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
