## [Reviewer Report]

I would like to commend the authors for undertaking this important and timely review on interventions aimed at improving mental health help-seeking attitudes, intentions, and behaviours. The topic is highly relevant to global mental health research and practice, particularly given the ongoing challenges in bridging the care and treatment gap worldwide and the increased attention on improving the supply side of this gap.

While the manuscript is valuable, I believe it would benefit from clearer definitions and some restructuring of the write-up to enhance its overall quality and readability. My detailed comments are provided below.

Introduction

- Line 75: Please check the capital “H” in “help-seeking.”

- This review represents a strong attempt to identify available evidence on interventions addressing the entire help-seeking process. It commendably includes all outcomes in the process, across any population, type of help, and context or country. While the authors briefly explain the help-seeking process and the Theory of Planned Behaviour (TPB), it would be useful to clarify the rationale / importance of focusing on all these elements (e.g. to have population-level impact, attitudes among general population are important, but also actual help-seeking intentions and behaviours among those experiencing mental health conditions).

- If space allows, I recommend briefly describing why the term help-seeking is a complex construct, specifically, that in most studies it has not been clearly defined (e.g., which part of the process or which source of help is being targeted) and that there is no consensus on its definition or measurement. The recommendations on what to specify in this type of research, by e.g. Rickwood & Thomas, 2014, would be helpful to reference in this context. This could also be used as justification for the broad focus of this review, and challenges with heterogeneity that are mentioned in the limitations section.

- It is helpful that the authors aim to update the previously conducted systematic review and meta-analysis by Xu et al.; however, as the methods differ considerably, I recommend mentioning this and removing direct comparisons with Xu et al. as they were able to conduct a meta-analysis.

Methods

- In the Methods section, it would be beneficial to specify the type of help included, the outcome measures, and the timeframe of outcome measures (following the recommendations by Rickwood & Thomas, 2014. This will be important for the interpretation of the results, for instance, some studies measure behaviour based on self-reflection, others based on service records, and some studies assess help-seeking at any point in the participant’s lifetime while others during a 3-month period.

- It would be good to add a reference for the definitions of the three outcomes presented, and for behaviour, the current definition refers to actual observable behaviour; however, as several studies use self-reported measures, broadening this definition may be appropriate.

- Please describe the method used to analyse the content of interventions and how these were mapped onto Table 2.

Results

- Table 1: I recommend including the country in which each study took place. Apart from a reference on line 173, the reader cannot determine which strategies were used in specific contexts. This information would support the practical uptake of the review results.

- Table 1: Please include, within the outcome measures, the type of help that was sought (in some cases, this is provided, but in others it is not).

- Table 1: It can be confusing to see that the ATSPPH was used to measure intentions. I see that the actual results are reported in terms of intentions, so I understand the choice, but it would be helpful to reflect on this somewhere in the paper.

- Table 1: Please check the references, e.g. Casanas et al. does not appear to be listed in the reference section.

- Overall, the narrative description of the results was somewhat challenging to follow. Including percentages or a summary table of study characteristics would improve clarity i believe.

- Similarly, in the section starting at line 235, it is not entirely clear which components appear to be most promising (e.g., improving help-seeking behaviour in schools). The authors could consider highlighting only the effective interventions and providing a short description of their content. This would improve clarity and help demonstrate which components were most effective, for which outcomes and settings. I would then also add the country in which the study took place, as that will be of interest to the reader of this journal.

- Line 223: The term “triangulation” may not be accurate in this context, as it seems to refer to combining results or content from the same source.

- It would also be helpful to explain the intervention components more clearly. For instance, the difference between “mental health literacy and psychoeducation” and “providing information about support services and treatment options” is not immediately apparent.

Discussion

- The discussion would benefit from a clear summary of which interventions were effective for which outcomes and populations (e.g. general population vs. those in need of help, child vs adult). Reflecting on which strategies might have population-level impact would also be valuable.

- It would be interesting to see whether any findings relate to the intention–behaviour gap, as previous literature (e.g., Rickwood & Thomas, 2014) suggests mixed evidence on the strength of this association.

- Line 381: The claim that these interventions are cost-effective and scalable may need a supporting explanation.

- It would strengthen the discussion to reflect on whether context seem to have influenced the effectiveness of components (e.g., which approaches were successful in low-resource settings or LMICs). Readers of this journal would likely be particularly interested in this aspect.

Overall, this manuscript offers a valuable and timely synthesis of interventions to enhance mental health help-seeking attitudes, intentions, and behaviours. I appreciate the authors’ thoughtful work and hope the feedback above helps strengthen the paper for publication.

---

## [Reviewer Report]

The manuscript presents a well-written systematic review of the current evidence regarding interventions aimed at improving help-seeking attitudes, intentions, and behaviors. The review adheres to the PRISMA guidelines, and the screening process is conducted with high methodological rigor. Both the research questions and the corresponding search strategy are appropriate and well-aligned.

However, several issues warrant further attention or clarification within the manuscript:

1. PRISMA Guidelines: The manuscript references the PRISMA guidelines and checklist from 2009, but it should be updated to the most recent version, the PRISMA 2020 guidelines and checklist.

2. Inclusion of Studies Published After 2016: The rationale for restricting the review to studies published after 2016 is unclear. This narrow time frame appears somewhat unusual, and it is questionable whether excluding earlier studies contributes meaningfully to advancing our understanding of the topic.

3. Exclusion Criteria: More detail is needed regarding the exclusion criteria during the abstract screening process. The manuscript provides limited information on the specific reasons for excluding a substantial number of publications.

4. Table 1: While Table 1 is useful, I strongly recommend including additional details, such as sample size and the male-to-female ratio. These data are essential for evaluating the effects reported in the studies and providing a clearer understanding of the sample demographics.

5. Meta-Analysis: I respectfully disagree with the authors' decision to refrain from conducting meta-analyses due to the heterogeneity of the interventions. Meta-analyses have been successfully conducted with smaller numbers of randomized controlled trials (RCTs) exhibiting similar levels of heterogeneity in interventions and outcomes.

6. Standardized Effect Sizes: Although I respect the authors’ reasoning for not conducting a meta-analysis, I suggest that they consider providing standardized effect sizes for the respective studies to facilitate some kind of comparison. P-values, as presented, are often highly dependent on sample size and may not be the most informative measure of intervention effectiveness.

7. Reorganization of Intervention Information: The sections discussing interventions across different settings could benefit from reorganization into a table or clearer structure. As it stands, the current presentation is difficult to follow and hampers readability.

8. Discussion: The discussion adequately summarizes the findings. Overall, the paper is well-written and accessible, with the exceptions outlined above.

---

## [Editor Report]

Thank you for submitting your review article to Cambridge Prisms: Global Mental Health. The topic is very timely. Kindly attend to the recommendations for strengthening the article recommended by reviewers one and three and resubmit.

---

## [Reviewer Report]

The authors have done a good job in revision the manuscript. I understand the rationale for not conducting a meta-analysis in their case although I still think there may be ways to do it. For the systematic review, I also accept the rational for not including papers before 2016, when the last systematic review was published. However, going back to the idea of a meta-analysis, using all studies might provide an opportunity to have sufficient numbers to conduct meaningful meta regression and provide insights into heterogeneity.

I am not entirely happy with the decision to not even calculate effect sizes for the studies included, when not provided but possible to calculate. I understand this is some more work, but this would at least allow the reader some quantitative comparison. So I am asking the authors to invest some more efforts into this.